# REDOUBT: Duo Safety Validation for Autonomous Vehicle Motion Planning

**Shuguang Wang**[1]  **Qian Zhou**[1]  **Kui Wu**[2]  **Dapeng Wu**[1]  **Wei-Bin Lee**[3]  **Jianping Wang**[1]

[1]City University of Hong Kong, Hong Kong, China
[2]University of Victoria, B.C., Canada
[3]Information Security Center, Hon Hai Research Institute, Taipei, Taiwan
`sgwang6-c@my.cityu.edu.hk`
`{qiazhou, jianwang}@cityu.edu.hk`  `wkui@uvic.ca`
`dpwu@ieee.org`  `wei-bin.lee@foxconn.com`

## Abstract

Safety validation, which assesses the safety of an autonomous system's motion planning decisions, is critical for the safe deployment of autonomous vehicles. Existing input validation techniques from other machine learning domains, such as image classification, face unique challenges in motion planning due to its contextual properties, including complex inputs and one-to-many mapping. Furthermore, current output validation methods in autonomous driving primarily focus on open-loop trajectory prediction, which is ill-suited for the closed-loop nature of motion planning. We introduce REDOUBT, the first systematic safety validation framework for autonomous vehicle motion planning that employs a duo mechanism, simultaneously inspecting input distributions and output uncertainty. REDOUBT identifies previously overlooked unsafe modes arising from the interplay of In-Distribution/Out-of-Distribution (OOD) scenarios and certain/uncertain planning decisions. We develop specialized solutions for both OOD detection via latent flow matching and decision uncertainty estimation via an energy-based approach. Our extensive experiments demonstrate that both modules outperform existing approaches, under both open-loop and closed-loop evaluation settings. Our codes are available at: `https://github.com/sgNicola/Redoubt`.

## 1   Introduction

Motion planning—the process of generating a trajectory for the ego-vehicle based on sensed traffic scenario data—is a cornerstone of autonomous driving. Learning-based motion planning methods [48, 54, 47, 45, 7], which leverage datasets of human driving behavior to train autonomous systems, have gained increasing attention due to their potential for greater adaptability compared to traditional rule-based approaches. Despite significant advances, autonomous vehicles continue to struggle with the full complexity of real-world driving scenarios [11, 29, 28, 42].

Safety validation for motion planning, which assesses whether a vehicle's decisions are safe in a given scenario, is thus critical. When an unsafe decision is detected, the validation module can: (1) immediately override the decision and initiate fallback measures (e.g., manual takeover [1]) to ensure short-term safety, and (2) flag the scenario as "challenging" to prioritize improvement in subsequent continual learning iterations [37, 41]. Safety validation could reduce or prevent tragedies like the March 29, 2025, Xiaomi SU7 incident, which resulted in three fatalities [12]. Nevertheless, such a systematic safety validation framework is currently lacking—a gap our this work seeks to address.

Safety validation can potentially be approached via two main angles: *input inspection* and *output inspection*. In the first category, *input inspection*, we draw inspiration from *Out-of-Distribution*

39th Conference on Neural Information Processing Systems (NeurIPS 2025).

*(OOD) detection* in other machine learning fields, such as image classification [49, 3], to establish a novel approach for assessing safety levels in autonomous driving. This approach determines whether a given input falls within the trained model's familiar data distribution: it is termed In-Distribution (InD) if it does, and otherwise Out-of-Distribution (OOD) [51]. In our context, if a traffic scenario is identified as OOD, the system could downgrade the safety score of the resulting decision. However, *input inspection* strategies based on OOD detection have not yet been used in motion planning, and adapting them poses unique challenges. Unlike classification tasks, motion planning requires predicting distributions over plausible future trajectories conditioned on complex, context-dependent inputs [52]. This one-to-many mapping obscures the underlying data distribution, making it difficult to define meaningful feature distances or confidence margins for OOD detection.

In the second category, *output inspection*, safety validation assesses the *uncertainty* of the autonomous system's output and deems those with high uncertainty as unsafe. However, these approaches are mainly designed for trajectory prediction rather than motion planning [35, 16]. In a nutshell, trajectory prediction predicts the future paths of other road agents, e.g., vehicles, pedestrians, and cyclists, based on their current/past behavior and environmental context, while motion planning determines the ego vehicle's own trajectory to navigate safely and efficiently toward a goal, considering predicted trajectories of others. Safety validation for trajectory prediction faces significant barriers when applied to motion planning. This is because trajectory prediction is typically validated in an open-loop setting [39, 36], where predicted trajectories of surrounding agents are compared against ground-truth logs from recorded datasets. In contrast, motion planning operates in a closed-loop manner: the ego-vehicle continuously uses feedback from its current state and the dynamic environment to update and adjust its plans in real time [17]. Consequently, applying existing *output inspection* approaches directly to motion planning could lead to critical failures, such as falsely certifying the ego vehicle's trajectory as safe when it actually results in collisions or road departures [10].

We propose **REDOUBT**, the first framework to systematically integrate *input inspection* through OOD detection and *output inspection* via uncertainty estimation for the safety validation of autonomous vehicle motion planning.

First, we develop specialized solutions for OOD scenario detection and uncertain decision estimation within the context of autonomous vehicle motion planning. We identify distributional shifts by estimating distribution likelihoods in the latent space using flow matching. Specifically, we employ a velocity field, enhanced with Gaussian Fourier Projection to capture temporal dynamics, to match target distributions and enable a principled estimation of distributional shifts in trajectory evolution. For estimating planning decision uncertainty under closed-loop dynamics, we introduce an energy-based approach that assigns risk-sensitive energy scores. These scores, computed based on future distances to obstacles or surrounding agents, provide a dense and continuous measure of safety risk, unlike sparse binary violation labels. During inference, we aggregate energy scores and trajectory probabilities to assess safety risks under closed-loop execution.

Moreover, REDOUBT employs a duo validation approach—simultaneously inspecting input distributions (traffic scenarios) and output uncertainty (planning decisions). The key insight behind this duo strategy, supported by our experiments, is that input and output validation are *complementary*, not interchangeable. By jointly validating both dimensions, it categorizes driving situations into four types: (a) **InD scenario + certain decision** → *Safe*, (b) **InD scenario + uncertain decision** → *Unsafe*, (c) **OOD scenario + certain decision** → *Unsafe*, and (d) **OOD scenario + uncertain decision** → *Unsafe*. Notably, REDOUBT uniquely identifies situations (b) and (c), which have been overlooked by existing approaches that rely solely on input or output inspection, thereby offering more comprehensive safety guarantees. In summary, our contributions are:

- We propose a novel duo safety validation approach for autonomous driving motion planning that concurrently inspects both input and output, revealing two previously overlooked unsafe situations.

- We design a latent flow matching method to estimate distributional shifts and detect OOD traffic scenarios, effectively modeling the multi-modal and dynamic nature of planning contexts.

- We present an energy-based risk prediction approach that integrates risk signals and trajectory probabilities to estimate driving decision uncertainty without requiring explicit violation labels.

- We outperform previous approaches on the nuPlan dataset in both OOD detection and decision uncertainty estimation. Our method demonstrates superior performance in both open-loop and closed-loop settings.

## 2 Related Work

**Out-of-Distribution (OOD) Detection.** OOD detection has been extensively studied in image classification [32, 53, 50] but remains underexplored in motion planning. Post-hoc methods such as Maximum Softmax Probability [19], MaxLogit [18], and energy-based scoring [34] derive OOD scores from the output layer of trained classifiers, offering simplicity and ease of implementation. Feature-space methods, including Mahalanobis distance [26] and k-Nearest Neighbor (KNN) approaches [40], estimate distributional deviation based on distances in the learned representation space. However, applying these methods to motion planning is challenging—post-hoc scores often yield overconfident predictions, while feature-space methods struggle with the one-to-many mapping and complex inputs, which obscure the data distribution.

**Motion Planning.** Motion planning aims to generate safe and efficient trajectories for autonomous agents. Hybrid methods, such as GC-PGP [10], integrate learning-based models with rule-based priors through centerline selection algorithms, achieving a balance between adaptability and safety. Fully learning-based models, including GameFormer [22] and PlanTF [8], leverage scene encoding to learn decision-making policies, generate multi-modal trajectory candidates, and select optimal plans based on trajectory probabilities and constraint satisfaction. Methods such as PLUTO [7] address distribution shift via contrastive learning, using data augmentations to shape latent representations. PlanScope [46] incorporates auxiliary losses based on Euclidean Signed Distance Fields (ESDF) to model cost and enforce safety constraints during planning. While these approaches have significantly advanced the performance of learning-based motion planning, they primarily overlook OOD detection.

**Uncertainty Estimation.** Uncertainty estimation in motion prediction has emerged as a valuable tool for safety validation of trajectory forecasts [36]. For example, Joodu [44] uses imitation error as a proxy for uncertainty, while Filos et al.[13] leverage disagreement among predicted trajectories to detect potential distribution shifts. DECODE [27] employs normalizing flows for domain awareness and selects the model with the highest likelihood to generate trajectory predictions. SOTIF [38] measures perceptual uncertainty with entropy, providing signals that guide risk evaluation in downstream motion prediction. Other works incorporate uncertainty arising from map inaccuracies and occlusions into trajectory prediction[16], and utilize trajectory distribution entropy to quantify predictive uncertainty [35]. Despite their utility, these uncertainty estimation approaches remain largely restricted to open-loop settings, relying on surrogate metrics such as imitation loss. These metrics fail to capture critical closed-loop risks—including collisions and road departures.

## 3 Methodology

This section introduces the REDOUBT framework, which consists of two key modules (dashed boxes in Figure 1): (1) an Out-of-Distribution (OOD) Detection module based on Flow Matching and (2) a Decision Uncertainty Estimation module. REDOUBT is orthogonal to and can be integrated with various existing motion planning solutions to validate the safety of their planned trajectories.

### 3.1 Learning-based Motion Planning

To begin, we outline the background of modern ML-based motion planning models. These models leverage scene encoders to transform multi-modal driving contexts into high-dimensional latent representations [23, 7], which encapsulate semantic and contextual information essential for trajectory prediction [25, 24, 33]. Formally, the driving context is encoded as a scene representation $\mathbf{x} = \{E_E, E_A, E_O, E_M\}$, where $E_E$ denotes the ego vehicle state, $E_A$ the states of dynamic agents, $E_O$ static object features, and $E_M$ map information. The model predicts future trajectories for dynamic agents, denoted as $Y_{1:N_d}^{1:T_f} = \{y_1^{1:T_f}, \ldots, y_{N_d}^{1:T_f}\}$. Furthermore, it generates a set of $N_m$ multi-modal ego trajectories, represented as $Y_0^{1:T_f} = \{(y_{0,i}^{1:T_f}, p_i)\}_{i=1}^{N_m}$ where $p_i$ is the probability associated with the $i$-th candidate trajectory. The final trajectory $\tau^*$ is selected by optimizing $p_i$ while adhering to traffic-related constraints. Building on this, we propose a latent-space-driven framework for OOD detection and decision uncertainty estimation.

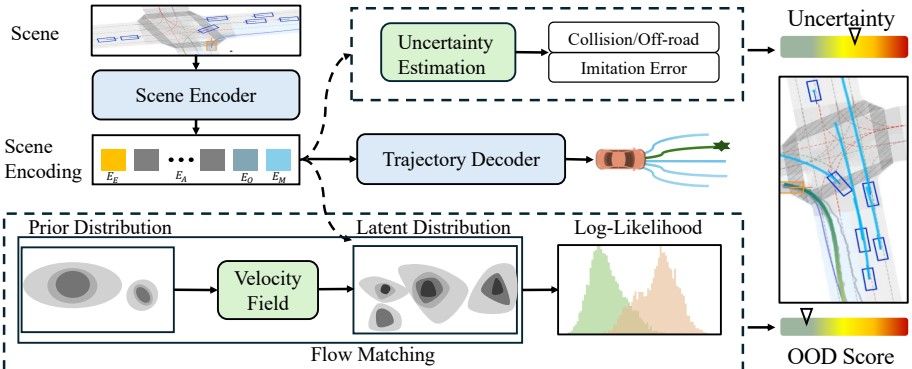

Figure 1: Overview of the proposed REDOUBT framework. The OOD Detection module utilizes Flow Matching to model the In-Distribution density of the latent space, learning a velocity field to transform the prior known distribution into the target distribution. The Decision Uncertainty Estimation module predicts collision and off-road violations, as well as imitation errors. The final decision-making process considers both the OOD score and the decision uncertainty.

## 3.2 OOD Detection based on Flow Matching

The OOD Detection module is designed to identify "unfamiliar" input scenarios and trigger a signal of potential danger. Similar to OOD detection approaches in other domains (e.g., [15, 53]), it can be described as employing a score function to determine whether a given input belongs to the In-Distribution (InD) or the Out-of-Distribution (OOD). We derive this score function from advancements in the generative AI field, applying it for safety validation. The core intuition is that generative models create a score to evaluate the similarity between the current distribution and the target distribution, guiding their path toward the target. Leveraging this concept in a novel way, we use the score to measure an input traffic scenario's proximity to the InD density.

**Preliminary on Flow Matching.** Flow Matching (FM) is a recently proposed paradigm for training generative models by transforming a simple source distribution into a target data distribution [30, 31, 9]. The aim of FM is to learn a time-dependent flow $\phi_t$ that continuously maps samples from a known source distribution, typically a standard Gaussian, to the target distribution [2]. This flow $\phi_t : [0, 1] \times \mathbb{R}^d \to \mathbb{R}^d$ is defined by an ordinary differential equation (ODE) parameterized by a velocity field $u_t$:

$$\frac{\mathrm{d}\phi_t(\mathbf{x})}{\mathrm{d}t} = u_t(\phi_t(\mathbf{x})). \tag{1}$$

The time-dependent probability path $(p_t)_{0 \leq t \leq 1}$, which smoothly interpolates between the source distribution $p_0 = p$ and the target distribution $p_1 = q$. A velocity field $u_t$ is suggested to generate this probability path. To learn the velocity field, a neural network $u_t^\theta$ is used to approximate the target velocity field $u_t$. The parameters $\theta$ of the neural network are typically estimated by minimizing the following objective function:

$$\mathcal{L}_{FM}(\theta) = \mathbb{E}_{t \sim U[0,1], \mathbf{x} \sim p_t} \|u_t^\theta(\mathbf{x}) - u_t(\mathbf{x})\|^2, \tag{2}$$

where $u_t(\mathbf{x})$ represents the target velocity field derived from the continuity equation.

**Velocity Field.** Let $\mathbf{z} \in \mathbb{R}^d$ represent the latent encoding of a motion planning trajectory obtained from the scene encoder. We define a time-dependent flow $\phi_t(\mathbf{z})$, governed by a neural velocity field $u_t^\theta$, which evolves along an interpolation path from a known prior distribution $p_0$ to the target data distribution $p_1$. To incorporate the temporal dimension, we employ a Gaussian Fourier Projection to embed the scalar time $t$ into a high-dimensional feature vector. The time embedding is given by:

$$\gamma(t) = [\sin(2\pi t W), \cos(2\pi t W)], \quad W \in \mathbb{R}^{d_t/2}, \tag{3}$$

where $W$ is a projection matrix that encodes multi-frequency information. This time embedding is then processed by a residual Multi-Layer Perceptron (MLP) architecture.

The velocity field is trained using a velocity matching loss, which minimizes the discrepancy between the predicted flow velocity $u_t^\theta(\mathbf{z}_t)$ and the velocity $\frac{d\mathbf{z}_t}{dt}$ along the interpolation path. The loss function is given by:

$$\mathcal{L}_{\text{flow}} = \mathbb{E}_{t \sim U[0,1], (\mathbf{z}_0, \mathbf{z}_1) \sim \pi_{0,1}} \left[ \left\| u_t^\theta(\mathbf{z}_t) - \frac{d\mathbf{z}_t}{dt} \right\|^2 \right], \tag{4}$$

where $\mathbf{z}_t$ is sampled along an interpolation path between $\mathbf{z}_0$ and $\mathbf{z}_1$, with velocity $\frac{d\mathbf{z}_t}{dt}$ along the path. Here, $\pi_{0,1}$ defines the joint distribution of $\mathbf{z}_0$ and $\mathbf{z}_1$, and the interpolation is guided by the Affine Probability Path under a Conditional Optimal Transport schedule [14].

**Computing Log-Likelihood.** Once the velocity field is trained, the log-likelihood of a sample $\mathbf{z}_1 \sim p_1$ can be computed by solving the flow ODE backward in time using an Euler solver. For the prior distribution $p_0$, we adopt a Gaussian Mixture Model (GMM) with $N_m$ components to represent the multi-modal latent space. Using the Instantaneous Change of Variables formula [6], the log-likelihood of the target distribution is expressed as:

$$\log p_1(\mathbf{z}_1) = \log \left( \frac{1}{N_m} \sum_{i=1}^{N_m} \mathcal{N}(\mathbf{z}_0 \mid \boldsymbol{\mu}_i, \sigma_i^2 \mathbf{I}) \right) - \int_0^1 \nabla \cdot u_t^\theta(\mathbf{z}_t)\, dt, \tag{5}$$

where $\mathcal{N}(\mathbf{z}_0 \mid \boldsymbol{\mu}_i, \sigma_i^2 \mathbf{I})$ represents the $i$-th Gaussian component of the prior distribution, characterized by mean $\boldsymbol{\mu}_i$ and isotropic covariance matrix $\sigma_i^2 \mathbf{I}$. The GMM contains $N_m$ components, each with an equal mixture weight of $1/N_m$. The term, $-\int_0^1 \nabla \cdot u_t^\theta(\mathbf{z}_t)\, dt$, is the divergence integral, which quantifies the density evolution as the sample flows from $\mathbf{z}_1$ to $\mathbf{z}_0$ under the velocity field $u_t^\theta(\mathbf{z}_t)$.

### 3.3 Decision Uncertainty Estimation

The Decision Uncertainty Estimation module is a multi-output network designed to predict various uncertainty metrics for planned trajectories, encompassing both open-loop and closed-loop measures. First, in open-loop settings, uncertainty is represented by imitation error, often quantified using Average Displacement Error (ADE): $\text{ADE} = \frac{1}{T_f} \sum_{t=1}^{T_f} \|y_t - \hat{y}_t\|_2$, where $y_t$ is the ground truth trajectory, and $\hat{y}_t$ is the predicted trajectory. Second, in closed-loop dynamics, uncertainty is indicated by two metrics: 1) No-Fault Collisions (NFC), which refer to collisions that could have been avoided with proper planning, and 2) Drivable Area Compliance (DAC), which assesses whether the ego vehicle operates within the mapped drivable area. Safety violations are represented as Boolean values (0: no violation, 1: violation). In the absence of violation labels during closed-loop no-feedback training, we propose a geometry-driven Violation Energy Score as pseudo-labels [43]. Leveraging signed distance as a physically intuitive metric, it transforms trajectories into continuous risk signals, capturing finer-grained risk features and addressing data scarcity in collision and driving area violation scenarios. The Violation Energy Score is further combined with trajectory probability to estimate decision uncertainty.

**Violation Energy Score.** At first, we construct the cost map using the Euclidean Signed Distance Field(ESDF) according to prior work [7, 54]. Non-drivable regions, such as obstacles and off-road areas, are rasterized into a binary mask. A distance transform is applied to compute a signed distance field, encoding the shortest distance of each pixel to the nearest obstacle or boundary. The vehicle is approximated as a set of $N$ covering circles, each with a predefined radius $R_c$. Each point in the trajectory $\tau = \{y^t \mid t = 1, \ldots, T_f\}$ is projected onto the ESDF, where bilinear interpolation is used to calculate the signed distance $d_i^t$ between the $i$-th covering circle at time $t$ and the nearest semantic region. A penalty is applied when $d_i^t < R_c$, enforcing adherence to drivable area constraints.

The violation energy for a given trajectory is defined as:

$$E = \frac{1}{T_f} \sum_{t=1}^{T_f} \sum_{i=1}^{N} \mathbf{M}_i^t \cdot \varphi(d_i^t), \tag{6}$$

where $T_f$ is the trajectory length, $N$ is the number of covering circles, $\mathbf{M}_i^t$ is a binary mask indicating whether the $i$-th circle at time $t$ intersects a region of interest (e.g., obstacles or boundaries), and $\varphi(d_i^t)$ is a penalty function quantifying the risk associated with the signed distance $d_i^t$.

To address different safety constraints, we define two penalty functions: $\varphi_{\mathrm{nfc}}(d_i^t)$ for No at-fault Collisions (NFC) and $\varphi_{\mathrm{dac}}(d_i^t)$ for Drivable Area Compliance (DAC). The NFC penalty is defined as:

$$\varphi_{\mathrm{nfc}}(d_i^t) = \max(0, \exp(R_c - d_i^t) - 1), \tag{7}$$

which grows exponentially as $d_i^t$ approaches or falls below $R_c$. This emphasizes the critical importance of avoiding overlaps with obstacles. When $d_i^t \geq R_c$, the penalty is zero, indicating a safe distance. This exponential design ensures heightened sensitivity near obstacles, particularly in dynamic or high-risk scenarios.

The DAC penalty is defined as:

$$\varphi_{\mathrm{dac}}(d_i^t) = \max(0, R_c - d_i^t), \tag{8}$$

which increases linearly as $d_i^t$ approaches the boundary ($d_i^t < R_c$). This formulation encourages smooth corrective actions and avoids over-penalizing minor deviations. For $d_i^t \geq R_c$, the penalty is zero, indicating compliance.

**Training Objective.** The $E_k(\tau)$ is transformed into a soft binary indicator by comparing it to a threshold $\epsilon$. The module predicts this signal using a likelihood prediction head, which outputs the logits $f_k(\tau)$. These logits are optimized using the Binary Cross-Entropy Loss with Logits, defined as:

$$\mathcal{L}_c = \sum_{k \in \{\mathrm{nfc,dac}\}} \mathrm{BCEWithLogits}(f_k(\tau), \mathbb{I}[E_k(\tau) > \epsilon]), \tag{9}$$

where $\mathbb{I}[E_k(\tau) > \epsilon]$ is the threshold indicator for each safety objective ($k \in \{\mathrm{nfc, dac}\}$), and $E_k(\tau)$ is the Violation Energy Score associated with the respective constraint. The total loss function combines the constraint loss $\mathcal{L}_c$ with an imitation loss $\mathcal{L}_{\mathrm{imitation}}$, which predicts the imitation error of the driven trajectory. The total loss is given by:

$$\mathcal{L}_{\mathrm{total}} = \mathcal{L}_{\mathrm{imitation}} + \mathcal{L}_c = \|\mathbf{e} - \hat{\mathbf{e}}\|^2 + \mathcal{L}_c, \tag{10}$$

where $\mathbf{e}$ is the ground-truth imitation error, and $\hat{\mathbf{e}}$ is the predicted imitation error. During inference, the module predicts $\hat{E}_{nfc} = \sigma(f_{\mathrm{nfc}}(\tau))$, the estimated probability of a collision violation, and $\hat{E}_{dac} = \sigma(f_{\mathrm{dac}}(\tau))$, the estimated probability of a drivable area compliance violation. Additionally, the predicted imitation error $\hat{\mathbf{e}}$ is output to estimate trajectory imitation performance.

## 3.4 Ensemble with Trajectory Probability

As mentioned in Section 3.1, motion planning generates $N_m$ candidate trajectories $Y_0^{1:T_f} = \{(y_{0,i}^{1:T_f}, p_i)\}_{i=1}^{N_m}$, where $p_i$ denotes the probability of the $i$-th candidate trajectory. The final trajectory $\tau^*$ is selected by maximizing $p_i$, subject to traffic and safety constraints. To align with the planner's decision-making, we combine the module-predicted score with trajectory probabilities. Instead of directly using softmax probabilities, according to [20], we adopt an energy-based score $E_p$ to reflect the planner's confidence in its planned trajectory $\tau^*$. The InD score quantifies a scenario's similarity with the training distribution and is defined as:

$$E_p = \log \sum_{i=1}^{N_m} \exp(p_i), \quad E_{InD} = \lambda \log p_1(\mathbf{z}_1) + (1 - \lambda)E_p, \tag{11}$$

where $\log p_1(\mathbf{z}_1)$ represents the normalized log-likelihood of the scene context from Flow Matching. A high $E_p$ indicates the planner is confident in its distribution of candidate trajectories.

For decision uncertainty estimation,

$$E_{nfc} = \lambda \hat{E}_{nfc} + (1 - \lambda)E_p, \quad E_{dac} = \lambda \hat{E}_{dac} + (1 - \lambda)E_p, \tag{12}$$

where $\hat{E}_{nfc}$ and $\hat{E}_{dac}$ are the predicted collision and compliance risks. Normalization ensures that all risk terms and $E_p$ are on a comparable scale. The hyperparameter $\lambda \in [0, 1]$ controls the relative weighting of the predicted violation risks versus the trajectory's confidence.

# 4 Experiments

We first outline our experimental settings. Next, we present the quantitative results for the OOD Detection module and the Decision Uncertainty Estimation module, respectively, followed by ablation studies. These results demonstrate that each module significantly outperforms existing methods, offering more reliable safety validation. Finally, we provide qualitative demonstrations showing that the duo validation approach REDOUBT, based on these two modules, effectively categorizes driving situations into four types. Notably, it uncovers two unsafe situations overlooked by existing works, achieving more comprehensive safety guarantees for autonomous vehicles.

## 4.1 Experimental Setups

**Dataset.** We conduct our evaluations using nuPlan [4], which is currently the most widely used and the only large-scale dataset for motion planning evaluation. This dataset has 73 unique and well-labeled scenario types, defined by low-level driving attributes. We partition the dataset based on scenario frequency, following a strategy similar to that used in Shifts [36]. Specifically, the top 50% most frequent scenario types—which account for 95% of the total data—are categorized as the In-Distribution (InD) set, while the remaining, less common scenarios constitute the Out-of-Distribution (OOD) set. For more details, please refer to Appendix A.1

Our evaluations are carried out across three planning evaluation modes defined by the nuPlan [4]: (1) **Open-loop (OP):** The evaluation is conducted using log replay. *Notably, open-loop tests have limited value for motion planning, which continuously interacts with environments. However, they remain relevant in scenarios with mainly static obstacles, so we include them for evaluation completeness.* (2) **Closed-loop Non-Reactive (CNR):** Both the ego vehicle and other agents can deviate from their original trajectories, but the agents are non-reactive. (3) **Closed-loop Reactive (CR):** Both the ego vehicle and other agents can deviate, and the agents are reactive to the ego vehicle's behavior.

**Planners.** Our REDOUBT framework can be universally applied to various motion planning solutions to validate the safety of their planning results. To evaluate its effectiveness and model-agnostic nature, we integrate REDOUBT into the latent spaces of four diverse learning-based motion planners introduced in Related Work 2: **PlanTF**, **PLUTO**, **GameFormer**, and **PlanScope**. For implementation details, please refer to Appendix A.2.

**Baselines.** REDOUBT is evaluated on: (1) OOD Detection and (2) Decision Uncertainty Estimation. For OOD Detection, we compare against post hoc methods (MSP [19], MaxLogit [18], Entropy [35], Energy [34]) and feature-space methods (Mahalanobis Distance (MDS) [26], KNN [40], lGMM [44]).

For decision uncertainty estimation, we evaluate in Closed-loop (CNR and CR) and Open-loop (OP) settings. Closed-loop metrics include (1) No At-Fault Collisions (NFC), assessing whether the ego vehicle avoids fault-based collisions, and (2) Drivable Area Compliance (DAC), checking trajectory adherence to drivable areas. In the OP setting, we compute Average Displacement Error (ADE) within a bounded region. NFC and DAC are compared with the Energy score method, while ADE is evaluated against Shifts [36] baselines: Model Averaging (MA) and VAR [13].

**Evaluation Metrics.** We evaluate OOD detection using: (1) the false positive rate (FPR95) of OOD samples when the true positive rate of InD samples is 95%, and (2) the area under the receiver operating characteristic curve (AUROC). For decision uncertainty estimation, we consider: (1) the false positive rate (FPR95) of (No At-Fault Collision)/(Driving Area Compliance) samples when the true positive rate of violation samples is 95%, and (2) the AUROC. For imitation error, we use the area under the error-retention curve (R-AUC), which tracks trajectory imitation error as ground-truth labels progressively replace predictions in descending order of predicted error.

## 4.2 Evaluation of OOD Detection

Our method consistently outperforms existing OOD detection approaches across all three evaluation modes. As shown in Table 1, which compares OOD detection methods trained with the same four motion planning models, our approach achieves notable improvements in both FPR95 reduction and AUROC increase in OP, CNR, and CR modes. These advancements stem from the effective density estimation of InD features. Moreover, our method demonstrates strong performance across various motion planners, highlighting its versatility and reliability under diverse evaluation conditions.

Table 1: Comparison with OOD detection methods trained with the four motion planning models across three evaluation modes on nuPlan dataset. ↑ indicates larger values are better and vice versa. The best result in each column is shown in **bolded**. OP: Open-loop evaluation mode. CNR: Closed-loop non-reactive evaluation mode. CR: Closed-loop reactive evaluation mode.

| Mode | Method | PlanTF | | PLUTO | | GameFormer | | PlanScope | | Average | |
|---|---|---|---|---|---|---|---|---|---|---|---|
| | | FPR95↓ | AUROC↑ | FPR95↓ | AUROC↑ | FPR95↓ | AUROC↑ | FPR95↓ | AUROC↑ | FPR95↓ | AUROC↑ |
| OP | MSP | 88.92 | 64.29 | 79.70 | 74.15 | 91.16 | 62.13 | 83.85 | 71.37 | 85.91 | 67.99 |
| | MaxLogit | 69.04 | 75.14 | 64.99 | 74.06 | 93.84 | 54.08 | 83.86 | 71.65 | 77.93 | 68.73 |
| | Entropy | 88.96 | 58.77 | 71.56 | 62.64 | 96.77 | 50.89 | 84.24 | 66.87 | 85.38 | 59.79 |
| | Energy | 67.77 | 75.93 | 65.17 | 76.50 | 87.43 | 55.56 | 73.74 | 74.17 | 73.53 | 70.54 |
| | MDS | 94.54 | 65.81 | 69.02 | 80.55 | 70.05 | 70.43 | 81.32 | 76.74 | 78.73 | 73.38 |
| | KNN | 89.31 | 66.87 | 71.05 | 79.03 | 62.66 | 80.24 | 84.44 | 76.70 | 76.87 | 75.71 |
| | lGMM | 94.54 | 65.81 | 66.30 | 79.51 | 87.28 | 70.72 | 72.38 | 79.87 | 80.13 | 73.98 |
| | REDOUBT(Ours) | **51.02** | **84.48** | **53.54** | **85.50** | **59.59** | **84.82** | **56.18** | **85.83** | **55.08** | **85.16** |
| CNR | MSP | 91.22 | 59.50 | 80.94 | 73.56 | 91.60 | 59.65 | 84.23 | 70.68 | 87.00 | 65.85 |
| | MaxLogit | 77.79 | 69.68 | 66.84 | 78.24 | 92.81 | 52.71 | 76.53 | 74.61 | 78.49 | 68.81 |
| | Entropy | 92.21 | 55.83 | 75.17 | 62.79 | 95.96 | 51.39 | 82.07 | 65.65 | 86.35 | 58.91 |
| | Energy | 76.75 | 70.19 | 65.44 | 76.55 | 92.81 | 52.72 | 74.64 | 74.41 | 77.41 | 68.47 |
| | MDS | 92.39 | 65.45 | 63.49 | 81.50 | 68.74 | 71.68 | 69.39 | 80.09 | 73.50 | 74.68 |
| | KNN | 90.56 | 66.05 | 74.99 | 78.46 | **65.80** | 77.58 | 75.85 | 78.10 | 76.80 | 75.05 |
| | lGMM | 88.77 | 67.10 | 76.51 | 77.83 | 68.88 | 77.26 | 67.89 | 79.93 | 75.51 | 75.53 |
| | REDOUBT(Ours) | **65.11** | **81.41** | **57.11** | **84.83** | 66.35 | **80.22** | **50.10** | **85.54** | **59.67** | **83.00** |
| CR | MSP | 88.03 | 58.26 | 84.19 | 69.20 | 94.94 | 51.94 | 85.99 | 65.52 | 88.29 | 61.23 |
| | MaxLogit | 81.79 | 64.98 | 71.93 | 72.83 | 91.77 | 58.47 | 74.96 | 70.67 | 80.11 | 66.74 |
| | Entropy | 93.29 | 51.17 | 75.17 | 62.79 | 94.00 | 53.35 | 83.11 | 65.27 | 86.39 | 58.15 |
| | Energy | 81.25 | 65.36 | 71.38 | 70.36 | 93.84 | 54.08 | 73.25 | 71.25 | 79.93 | 65.26 |
| | MDS | 90.84 | 69.52 | 68.13 | 79.83 | 68.81 | 71.47 | 70.81 | 73.71 | 74.65 | 73.63 |
| | KNN | 90.01 | 66.14 | 73.23 | 78.66 | 67.17 | 77.01 | 67.00 | 80.82 | 74.35 | 75.66 |
| | lGMM | 86.31 | 71.74 | 74.41 | 77.37 | 69.33 | 76.63 | 73.34 | 76.96 | 75.85 | 75.68 |
| | REDOUBT(Ours) | **61.81** | **81.81** | **59.14** | **82.58** | **66.83** | **80.32** | **58.12** | **83.47** | **61.48** | **82.05** |

Figure 2: Retention curves of uncertainty estimation in open-loop evaluation across different planners. The ADE is plotted over the retention fraction. Lower R-AUC values indicate better performance. The oracle retention curve (green) is obtained by sorting samples in descending order of true ADE.

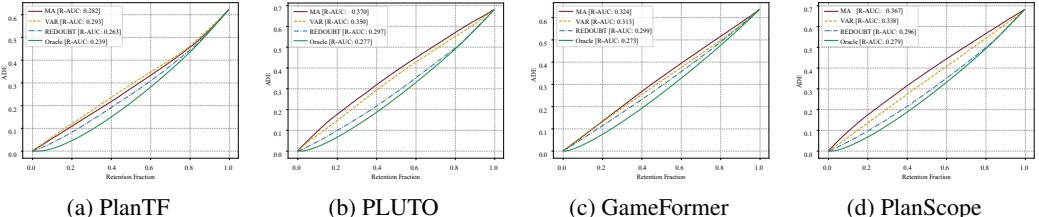

(a) PlanTF  (b) PLUTO  (c) GameFormer  (d) PlanScope

## 4.3 Evaluation of Decision Uncertainty Estimation

**Closed-loop Evaluation.** Table 2 presents the evaluation results of uncertainty estimation, including No At-fault Collision (NFC) and Driving Area Compliance (DAC) under the closed-loop non-reactive and closed-loop reactive modes. Our method demonstrates superior performance across all four motion planners. For example, for predicting NFC violations, our method improves the average AUROC by 12.81% and 10.03% in the CNR mode and the CR mode, respectively.

Table 2: Comparison of NFC and DAC estimation results, trained with the four motion planning models and evaluated under the CNR and CR modes on nuPlan dataset.

| Mode | Uncertainty | Method | PlanTF | | PLUTO | | GameFormer | | PlanScope | | Average | |
|---|---|---|---|---|---|---|---|---|---|---|---|---|
| | | | FPR95↓ | AUROC↑ | FPR95↓ | AUROC↑ | FPR95↓ | AUROC↑ | FPR95↓ | AUROC↑ | FPR95↓ | AUROC↑ |
| CNR | NFC | Energy | 79.33 | 63.10 | 89.52 | 66.36 | 91.63 | 52.93 | 73.40 | 57.30 | 83.47 | 59.92 |
| | | REDOUBT(Ours) | **70.93** | **70.45** | **68.04** | **71.62** | **56.85** | **74.44** | **57.30** | **74.41** | **63.28** | **72.73** |
| | DAC | Energy | 76.88 | 59.01 | 76.70 | 63.31 | 95.38 | 51.19 | 69.36 | 66.21 | 79.58 | 59.93 |
| | | REDOUBT(Ours) | **55.13** | **82.33** | **52.97** | **77.33** | **52.36** | **75.38** | **55.94** | **75.00** | **54.10** | **77.51** |
| CR | NFC | Energy | 82.74 | 64.39 | 82.27 | 71.58 | 87.88 | 59.62 | 80.84 | 67.24 | 83.43 | 65.71 |
| | | REDOUBT(Ours) | **67.04** | **71.97** | **58.29** | **78.11** | **57.34** | **73.37** | **49.22** | **79.51** | **57.97** | **75.74** |
| | DAC | Energy | 66.98 | 70.88 | 80.26 | 55.73 | 80.53 | 67.95 | 69.67 | 73.70 | 74.36 | 67.07 |
| | | REDOUBT(Ours) | **52.24** | **84.03** | **58.59** | **76.90** | **51.63** | **76.90** | **48.67** | **80.00** | **52.78** | **79.46** |

**Open-loop Evaluation.** The retention curves of uncertainty estimation under the open-loop evaluation mode, as shown in Figure 2, demonstrate that our method consistently performs closer to the Oracle retention compared to the best baselines across multiple planners. For example, in the PlanTF planner,

our method achieves an ADE R-AUC of 0.263, which is closer to the Oracle value of 0.239, compared to the best baseline (0.282). This indicates that our method exhibits better uncertainty calibration, as it effectively aligns uncertainty estimates with the true ADE.

## 4.4 Evaluation of Duo Safety Validation

We evaluate the full REDOUBT system under a closed-loop reactive setting, jointly using OOD detection and decision uncertainty estimation. We report recall as the primary metric, measuring coverage of potential unsafe scenarios. Here, potential unsafe scenarios comprise three categories: InD + Uncertain (InD inputs with driving violations), OOD + Uncertain (OOD inputs with driving violations), and OOD + Certain (OOD inputs without driving violations). Further qualitative analysis of unsafe scenarios is provided in Section 4.6.

Table 3: Duo safety validation under closed-loop reactive mode: recall across OOD detection and decision uncertainty estimation

| Method | PlanTF | PLUTO | GameFormer | PlanScope | Avg. |
|---|---|---|---|---|---|
| lGMM | 53.50 | 65.88 | 51.81 | 49.95 | 55.29 |
| Energy | 59.52 | 65.82 | 64.97 | 72.61 | 65.73 |
| **REDOUBT** | **90.24** | **90.52** | **91.34** | **90.02** | **90.53** |
| Category | Decomposition of Recall by Scenario Category | | | | |
| InD+Uncertain | 0.70 | 1.26 | 1.79 | 0.87 | 1.16 |
| OOD+Certain | 10.04 | 15.61 | 10.50 | 6.78 | 10.73 |
| OOD+Uncertain | 79.50 | 73.65 | 79.05 | 82.37 | 78.64 |

REDOUBT attains 90.53% recall, substantially outperforming strong baselines—55.29% for driving uncertainty estimation and 65.73% for OOD detection alone. This indicates that the unified design captures a broader set of unsafe cases.

We further decompose recall by category. OOD + Uncertain constitutes the majority (78.64%). InD + Uncertain accounts for 1.16%; despite its small share, these cases are safety critical. OOD + Certain contributes an additional 10.73%, reflecting OOD situations that may not immediately cause violations but remain hazardous under distribution shift. These results underscore the value of unifying OOD detection with decision uncertainty estimation for safety validation.

## 4.5 Ablation Studies

We conduct ablation experiments to evaluate the importance of the component design in our method.

**Impact of Trajectory Probability Energy.** We analyze the impact of incorporating the trajectory probability energy ($E_p$) on OOD detection and NFC and DAC estimation. Table 4 summarizes the average evaluation results without (w/o) and with $E_p$ (REDOUBT). Notably, the inclusion of $E_p$ significantly enhances OOD detection under all modes and NFC and DAC estimation under the CNR and CR modes. These findings demonstrate $E_p$'s effectiveness in improving OOD detection and uncertainty estimation, as it is enhanced by encoding trajectory probability information. Additionally, we analyze the sensitivity of the ensemble weight $\lambda$, which balances the contributions of flow matching and $E_p$ in Appendix B.4.

**Impact of Gaussian Fourier Projection.** Table 5 demonstrates the impact of the Gaussian

Table 4: Effects of Trajectory Probability Energy $E_p$ on OOD detection and Uncertainty Estimation.

| Metric | Method | OOD | | | NFC | | DAC | |
|---|---|---|---|---|---|---|---|---|
| | | OP | CNR | CR | CNR | CR | CNR | CR |
| FPR95 ↓ | w/o $E_p$ | 62.08 | 71.06 | 63.47 | 71.85 | 69.42 | 62.86 | 56.15 |
| | w/$E_p$ | 55.08 | 59.67 | 61.48 | 63.28 | 57.97 | 54.10 | 52.78 |
| AUROC ↑ | w/o $E_p$ | 83.23 | 78.47 | 80.82 | 68.18 | 72.12 | 77.02 | 78.58 |
| | w/$E_p$ | 85.16 | 83.00 | 82.05 | 72.73 | 75.74 | 77.51 | 79.46 |

Table 5: Effects of GFP module and GMM Prior.

| Mode | Module | FPR95 ↓ | AUROC ↑ |
|---|---|---|---|
| **OP** | w/o GFP | 61.67 | 82.88 |
| | w/o GMM | 61.11 | 84.42 |
| | **REDOUBT** | **55.08** | **85.16** |
| **CNR** | w/o GFP | 62.67 | 82.15 |
| | w/o GMM | 61.68 | 82.79 |
| | **REDOUBT** | **59.67** | **83.00** |
| **CR** | w/o GFP | 63.68 | 80.87 |
| | w/o GMM | 70.33 | 75.46 |
| | **REDOUBT** | **61.48** | **82.05** |

Fourier Projection (GFP) on OOD detection performance across three modes (OP, CNR, and CR). GFP, which embeds the scalar time $t$ into a high-dimensional feature vector, consistently improves AUROC scores. These results prove the effectiveness of GFP in capturing temporal features for better OOD detection. Further details are provided in Appendix B.2.

**Impact of Gaussian Mixture Model Prior.** The ablation study in Table 5 examines the impact of using a Gaussian Mixture Model (GMM) prior in flow matching, compared to a Standard Normal distribution. The results consistently demonstrate that the GMM prior enhances Out-of-Distribution (OOD) detection performance across all modes. Further details are provided in Appendix B.3.

## 4.6 Demonstration of Duo Safety Validation

REDOUBT simultaneously inspects input distributions and output uncertainty, categorizing driving situations into four combinations based on the interplay of InD/OOD scenarios and certain/uncertain planning decisions. See Figure 3 for examples of each category.

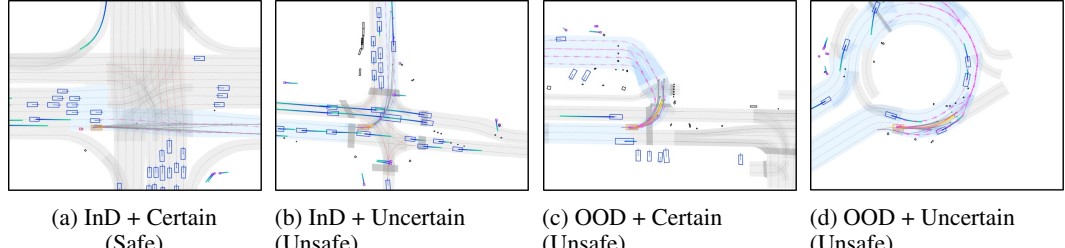

(a) InD + Certain  (b) InD + Uncertain  (c) OOD + Certain  (d) OOD + Uncertain
    (Safe)              (Unsafe)             (Unsafe)            (Unsafe)

Figure 3: Safety categorization in REDOUBT: (a) **InD + Certain** (Safe, e.g., straight intersection traversal); (b) **InD + Uncertain** (Unsafe, e.g., high-density left turns with trained-but-failed scenarios); (c) **OOD + Certain** (Unsafe, e.g., overconfident unprotected cross-turns); (d) **OOD + Uncertain** (Unsafe, e.g., exiting drivable areas at drop-off locations). The scenarios illustrated include: Orange lines for the planned trajectory of the motion planner, Blue lines for the predicted trajectories of agents, and Gray lines for the candidate trajectories by the planner.

REDOUBT identifies two previously overlooked unsafe modes and illustrates them with representative failures. First, the InD + Uncertain scenario is overlooked by an input-only inspection that treats all InD as safe. Even in familiar scenarios, inherent complexity can induce uncertain decisions and unsafe outcomes. In Case (b), the High-Density Traffic during Left Turns scenario involves a left-turn maneuver in a high-density traffic environment, which is a common challenge for motion planners. Although this category of scenario was encountered during training, we observed that the planned trajectory by PlanScope planner and the candidate trajectories still resulted in a collision with surrounding vehicles. This failure highlights the complexity of high-density interactions where subtle variations can lead to collisions despite prior training exposure.

Second, the OOD + Certain scenario is overlooked by output-only inspection that treats all "certain" decisions as safe. Rare scenarios can induce overconfident decisions that remain hazardous under shift. In Case (c), the Unprotected Cross-Turns scenario involves unprotected cross-turns, a category of scene that was not observed during training. Despite the absence of such scenarios in the training data, the planner still needs to output decisions. In Case (d), the Traversing Pickup/Drop-Off Areas scenario involves the ego vehicle traversing a drop-off area while not stopping. Such scenarios were also not present in the training data, and we observed that the planned trajectory drifted outside the drivable area. This failure illustrates the planner's limitations in handling complex, context-specific scenarios that require precise spatial reasoning and awareness.

## 5 Conclusion

In this work, we propose REDOUBT, a duo safety validation framework for autonomous vehicle motion planning that addresses two critical aspects: OOD detection and decision uncertainty estimation. Our approach introduces a latent flow matching method to model distributional shifts and detect OOD traffic scenarios, effectively capturing the planning contexts' multi-modal and dynamic nature. Additionally, we design an energy-based risk prediction method that integrates trajectory probabilities and risk signals to estimate decision uncertainty without relying on explicit violation labels. Experiments and ablation studies on the nuPlan dataset validate the effectiveness and versatility of REDOUBT, demonstrating its ability to identify previously overlooked unsafe situations.

## Acknowledgement

The work is supported in part by Hong Kong Research Grant Council under CRF C1042-23GF, GRF 11216323, and GRF 11214925.

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

# A More Experimental Details

## A.1 Training Settings

The training dataset contains a total of 300,000 scenarios, including specific InD scenario types. The evaluation dataset consists of 19,685 scenarios, with fixed scenario tokens. For each scenario type (except for the "Unknown" scenario, which cannot be verified based on the scenario type), either 500 scenarios are included, or, if fewer than 500 scenarios are available, all the available scenarios are used. All planners are trained exclusively on the InD dataset and evaluated on the full dataset, which includes both InD and OOD scenarios.

## A.2 Implementation Details

**Motion Planning Model.** Our experiments are conducted on a server with 8 NVIDIA RTX 5880 Ada GPUs and the Pytorch platform. The training experiments consist of two phases. In the first phase, each planner is trained according to its original settings. Specifically, PlanTF is trained with a batch size of 128, a weight decay of $1 \times 10^{-4}$, and for 25 epochs [8]. The learning rate starts at $1 \times 10^{-3}$ and decays to zero following a cosine schedule. PLUTO is also trained with a batch size of 128 and a weight decay of $1 \times 10^{-4}$ for 25 epochs [7]. Its learning rate is linearly increased to $1 \times 10^{-3}$ over the first 3 epochs and then follows a cosine decay. GameFormer is trained with a batch size of 32, a weight decay of 0.01, and for 20 epochs [22]. The learning rate starts at $1 \times 10^{-4}$ and halves every 2 epochs after the 10th epoch. PlanScope is trained with a batch size of 32 per GPU for 25 epochs, with 3 warm-up epochs, using the Adam optimizer and an initial learning rate of $1 \times 10^{-3}$ [46].

**REDOUBT Model.** In the second phase, the flow matching and uncertainty prediction modules are optimized separately. The flow matching module is trained for 2000 epochs using the Adam optimizer, a batch size of 1024, and a cosine learning rate schedule with an initial learning rate of $1 \times 10^{-3}$. The uncertainty prediction module is trained for 25 epochs, also using the Adam optimizer, a batch size of 1024, and a fixed learning rate of $1 \times 10^{-3}$. During this phase, the weights of the scene encoder and trajectory prediction decoder are frozen to preserve motion planning performance.

# B Additional Ablation Studies

Table 6: Detailed results of the ablation study on Trajectory Probability Energy $E_p$.

| Evaluation | Mode | Method | PlanTF | | PLUTO | | GameFormer | | PlanScope | | Average | |
|---|---|---|---|---|---|---|---|---|---|---|---|---|
| | | | FPR95 ↓ | AUROC ↑ | FPR95 ↓ | AUROC ↑ | FPR95 ↓ | AUROC ↑ | FPR95 ↓ | AUROC ↑ | FPR95 ↓ | AUROC ↑ |
| OOD | OP | w/o $E_p$ | 62.20 | 82.19 | 62.60 | 82.30 | 62.93 | 83.67 | 60.60 | 84.74 | 62.08 | 83.23 |
| | | w/ $E_p$ | **51.02** | **84.48** | **53.54** | **85.50** | **59.59** | **84.82** | **56.18** | **85.83** | **55.08** | **85.16** |
| | CNR | w/o $E_p$ | 90.48 | 69.10 | 65.35 | 81.52 | 66.83 | 79.92 | 61.58 | 83.32 | 71.06 | 78.47 |
| | | w/ $E_p$ | **65.11** | **81.41** | **57.11** | **84.83** | **66.35** | **80.22** | **50.10** | **85.54** | **59.67** | **83.00** |
| | CR | w/o $E_p$ | 65.77 | 80.43 | 65.84 | 80.40 | 66.89 | 79.64 | **55.38** | 82.82 | 63.47 | 80.82 |
| | | w/ $E_p$ | **61.81** | **81.81** | **59.14** | **82.58** | **66.83** | **80.32** | 58.12 | **83.47** | **61.48** | **82.05** |
| NFC | CNR | w/o $E_p$ | 74.78 | 69.01 | 71.68 | 68.11 | 76.62 | 62.96 | 64.30 | 72.62 | 71.85 | 68.18 |
| | | w/ $E_p$ | **70.93** | **70.45** | **68.04** | **71.62** | **56.85** | **74.44** | **57.30** | **74.41** | **63.28** | **72.73** |
| | CR | w/o $E_p$ | 73.31 | 69.55 | 72.90 | 76.77 | 59.12 | 71.08 | 72.34 | 71.09 | 69.42 | 72.12 |
| | | w/ $E_p$ | **67.04** | **71.97** | **58.29** | **78.11** | **57.34** | **73.37** | **49.22** | **79.51** | **57.97** | **75.74** |
| DAC | CNR | w/o $E_p$ | 55.72 | 82.28 | 71.86 | 75.53 | **52.35** | 75.33 | 71.49 | 74.94 | 62.86 | 77.02 |
| | | w/ $E_p$ | **55.13** | **82.33** | **52.97** | **77.33** | 52.36 | **75.38** | **55.94** | **75.00** | **54.10** | **77.51** |
| | CR | w/o $E_p$ | 53.55 | 83.55 | 69.02 | 76.69 | 52.12 | 75.58 | 49.90 | 78.50 | 56.15 | 78.58 |
| | | w/ $E_p$ | **52.24** | **84.03** | **58.59** | **76.90** | **51.63** | **76.90** | **48.67** | **80.00** | **52.78** | **79.46** |

## B.1 Impact of Trajectory Probability Energy

Table 6 compares performance with and without the inclusion of trajectory probability energy ($E_p$) across various models and evaluation scenarios. The results demonstrate that incorporating $E_p$ consistently enhances performance across all metrics (FPR95 and AUROC), as evidenced by lower FPR95 values and higher AUROC scores. The improvements are particularly significant in the OOD and NFC evaluations, where $E_p$ significantly reduces FPR95 and increases AUROC, especially for models such as PlanTF and PlanScope.

Table 7: Detailed results of the ablation study on Gaussian Fourier Projection (GFP).

| Mode | Module | PlanTF | | PLUTO | | GameFormer | | PlanScope | | Average | |
|------|--------|--------|--|-------|--|------------|--|-----------|--|---------|--|
| | | FPR95 ↓ | AUROC ↑ | FPR95 ↓ | AUROC ↑ | FPR95 ↓ | AUROC ↑ | FPR95 ↓ | AUROC ↑ | FPR95 ↓ | AUROC ↑ |
| OP | w/o GFP | 69.34 | 82.23 | 56.42 | 83.20 | 65.32 | 80.30 | **55.58** | 85.77 | 61.67 | 82.88 |
| | w/ GFP | **51.02** | **84.48** | **53.54** | **85.50** | **59.59** | **84.82** | 56.18 | **85.83** | **55.08** | **85.16** |
| CNR | w/o GFP | **63.88** | 80.73 | 62.87 | 82.15 | 66.69 | **80.23** | 57.25 | 85.49 | 62.67 | 82.15 |
| | w/ GFP | 65.11 | **81.41** | **57.11** | **84.83** | **66.35** | 80.22 | **50.10** | **85.54** | **59.67** | **83.00** |
| CR | w/o GFP | 65.74 | 79.10 | 65.20 | 81.54 | **56.85** | **80.72** | 66.94 | 82.13 | 63.68 | 80.87 |
| | w/ GFP | **61.81** | **81.81** | **59.14** | **82.58** | 66.83 | 80.32 | **58.12** | **83.47** | **61.48** | **82.05** |

## B.2 Impact of Gaussian Fourier Projection

Table 7 presents a comparison of OOD detection performance before and after removing the Gaussian Fourier Projection (GFP) in the velocity field module of flow matching. The results indicate that incorporating the GFP improves the overall OOD detection performance across all four planners.

Table 8: Detailed results of the ablation study on Gaussian Mixture Model (GMM).

| Mode | Module | PlanTF | | PLUTO | | GameFormer | | PlanScope | | Average | |
|------|--------|--------|--|-------|--|------------|--|-----------|--|---------|--|
| | | FPR95 ↓ | AUROC ↑ | FPR95 ↓ | AUROC ↑ | FPR95 ↓ | AUROC ↑ | FPR95 ↓ | AUROC ↑ | FPR95 ↓ | AUROC ↑ |
| OP | w/o GMM | 62.05 | 82.55 | 63.21 | **85.73** | 62.66 | 83.64 | 56.53 | 85.76 | 61.11 | 84.42 |
| | w/ GMM | **51.02** | **84.48** | **53.54** | 85.50 | **59.59** | **84.82** | 56.18 | **85.83** | **55.08** | **85.16** |
| CNR | w/o GMM | 67.56 | 81.08 | **55.72** | **85.31** | **62.76** | 80.17 | 60.66 | 84.59 | 61.68 | 82.79 |
| | w/ GMM | **65.11** | **81.41** | 57.11 | 84.83 | 66.35 | **80.22** | **50.10** | **85.54** | **59.67** | **83.00** |
| CR | w/o GMM | 75.45 | 73.64 | 71.38 | 73.97 | **62.80** | **80.36** | 71.70 | 73.86 | 70.33 | 75.46 |
| | w/ GMM | **61.81** | **81.81** | **59.14** | **82.58** | 66.83 | 80.32 | **58.12** | **83.47** | **61.48** | **82.05** |

## B.3 Impact of Gaussian Mixture Model Prior

Table 8 presents a comparison of OOD detection performance before and after replacing the Gaussian Mixture Model (GMM) with a standard distribution. The results indicate that GMM prior distribution is more effective.

Table 9: Inference time of REDOUBT on different motion planners.

| Planner | PlanTF | PLUTO | GameFormer | PlanScope |
|---------|--------|-------|------------|-----------|
| **Inf. Time** | 3.12 ms | 3.30 ms | 10.79 ms | 3.25 ms |

Figure 4: Ablation study on $\lambda$: (a) OOD Detection (OP Mode), (b) NFC Estimation (CNR Mode), and (c) DAC Estimation (CNR Mode).

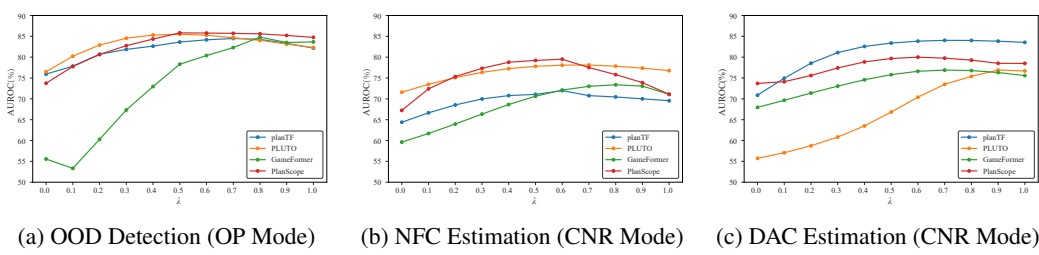

(a) OOD Detection (OP Mode)     (b) NFC Estimation (CNR Mode)     (c) DAC Estimation (CNR Mode)

## B.4 Hyperparameter Sensitivity

Figure 4 contains three subfigures illustrating performance trends for OOD detection under the OP mode, NFC estimation, and DAC estimation under the CNR mode. Each subfigure shows the

performance of various planners as $\lambda$ varies from 0 to 1, where $\lambda = 0$ represents the configuration without Flow Matching, and $\lambda = 1$ represents the configuration without $E_p$. The results indicate that the optimal range for $\lambda$ lies between 0.5 and 0.8. For example, OOD detection with $\lambda = 0.8$ achieves the optimal performance for GameFormer. The hyperparameter averages the strengths of both methods, avoiding the shortcomings of relying exclusively on one.

## C  Inference Time

Table 9 illustrates the inference time of REDOUBT on different motion planners. REDOUBT introduces an additional inference time of only 3.12 to 10.79 ms. Considering that the original planner operates at 20 Hz (resulting in a latency of 50 ms per frame), the extra latency from REDOUBT is minimal. Importantly, the total latency remains well below the maximum threshold of 1000 ms per inference mandated by the nuPlan benchmark [17]. These results demonstrate that REDOUBT is computationally efficient and capable of meeting the real-time performance requirements of modern motion planners.

## D  Limitations

We use nuPlan for validation because it is currently the only large-scale dataset specifically designed for motion planning, offering both open-loop and closed-loop evaluation settings. Most existing ML-based motion planners [22, 8, 46, 7]are exclusively trained and tested on nuPlan due to the lack of other comparable motion planning datasets. We recognize that further validation on additional datasets would enhance the generalizability of our framework, and we aim to include them in the future when new datasets become publicly available.

## E  Broader Impacts

This paper presents REDOUBT, a novel duo safety validation framework for motion planning in autonomous driving. Autonomous driving is a safety-critical domain where rigorous safety validation is crucial to prevent potentially hazardous decisions. REDOUBT is the first systematic safety validation framework that employs a duo mechanism to simultaneously inspect both input distributions and output uncertainty. This dual inspection approach reveals two previously overlooked unsafe scenarios arising from the interplay between InD/OOD inputs and certain/uncertain planning decisions.

To address these scenarios, we propose two specialized solutions: (1) OOD detection using latent flow matching, and (2) decision uncertainty estimation via an energy-based approach. Extensive experiments demonstrate that both modules significantly outperform existing methods under both open-loop and closed-loop evaluation settings. By improving the safety and reliability of autonomous driving systems, REDOUBT offers broad applicability across industries, contributing to safer and more dependable autonomous vehicles.

Notably, several leading autonomous driving companies, such as Momenta [5], have already adopted learning-based planners in their systems. Our method is well-suited for such deployments, as it integrates seamlessly into the pipeline. Specifically, the input to our method would be the latent space representation from the autonomous driving stack, and the output would include whether it represents an OOD scenario and risk probabilities of potential driving violations.

Several practical challenges need to be addressed for real-world deployment. Our approach could identify more unsafe scenarios than traditional methods. This raises the question of how to balance sensitivity and specificity in decision making [21]. To address these challenges, we propose that the output of our method serves as helpful, supplementary information to existing decision-making mechanisms within the stack. For instance, it could integrate with existing systems to trigger takeover requests as needed. Additionally, our system could be optionally enabled to prioritize safety in specific scenarios, such as high-risk environments or areas prone to OOD events.

