# OpenReview forum: "REDOUBT: Duo Safety Validation for Autonomous Vehicle Motion Planning"
_NeurIPS.cc/2025/Conference — NeurIPS 2025 poster_

### Official Review · Reviewer_pm9A · 2025-07-02

**Clarity:** 3
**Significance:** 2
**Originality:** 2
**Rating:** 4
**Confidence:** 4

**Summary:**

This paper introduces REDOUBT, a duo safety validation framework for motion planning task in autonomous driving. REDOUBT features more robust safety validation, through concurrent scenario OOD detection via latent flow matching, and decision uncertainty estimation quantified by ESDF and modal probability. REDOUBT is validated through multiple planners in both open and closed loop settings, proving effectiveness across other OOD detection methods.

**Questions:**

1.  How is the difficulty of a scenario quantitatively defined or evaluated? Can the method distinguish between hard-but-frequent vs. rare-but-trivial cases?

2. How does the proposed identification mechanism concretely benefit downstream planning performance, especially in closed-loop execution?

3. Since the separation of training and evaluation sets is manually based on frequency, is there a risk of implicit leakage or circular reasoning when evaluating OOD detection? How sensitive are the results to the chosen threshold of frequency?

4. To what extent do the detected OOD cases actually correlate with safety-critical planner failures? Could some OOD regions be benign or irrelevant to planning outcomes?

5. Is there an ablation showing the importance of each verification module (e.g., flow-based, energy-based)? Could a simpler or more modular combination of components achieve similar results?

**Ethical Concerns:**

["NO or VERY MINOR ethics concerns only"]

**Final Justification:**

After a detailed analysis through the rebuttal, the reviewer is convinced by the authors regarding the novelty in designing the safety validation framework. Still, its current evaluation is limited (as also noted by other reviewers) without close-loop planners. Hence, the reviewer is inclined to a borderline score.

**Limitations:**

yes

**Paper Formatting Concerns:**

N/A.

**Quality:**

3

**Strengths And Weaknesses:**

**Strengths**

1. The motivation for safety validation in autonomous driving is well-articulated and compelling.

2. The paper presents extensive comparisons with OOD detection baselines in both open-loop and closed-loop settings. REDOUBT demonstrates effectiveness across multiple widely adopted planning algorithms.

**Weaknesses**

1. The originality of each verification module is somewhat limited. Prior works such as [1] and [2] have also explored latent flow matching and energy-based methods for domain identification and uncertainty quantification.

2. Despite a thorough introduction and comprehensive experimental comparisons with OOD detection methods, the paper lacks deeper insights into how the proposed identification mechanism concretely improves or mitigates failures in open-/closed-loop planning.

3. Since the train/evaluation split is manually separated based on frequency, it is unclear whether the method can truly identify hard-but-frequent edge cases.

4. The qualiative results are not intuitative enough. More supplementary like closed-loop identification are expected. No code are provided for verification

5. How is each planner trained? Is there poential risk for data leakage?

[1] DECODE: Domain-aware Continual Domain Expansion for Motion Prediction

[2] SOTIF Entropy: Online SOTIF Risk Quantification and Mitigation for Autonomous Driving

---

> ### Author Rebuttal · Authors · 2025-07-31
>
> **Q1:The originality of the verification modules compared to prior works.**
>
> **A1:** Thanks for pointing out DECODE and SOTIF Entropy. We will incorporate a discussion into the **Related Work** section. Below, we outline the key differences:
>
> **1) Differences from DECODE：A. Flow Matching vs. Normalizing Flow:** While DECODE employs **Normalizing Flow (NF)**, we utilize **Flow Matching (FM)**. NF relies on maximum likelihood training of invertible transformations and requires computing the Jacobian determinant, a computationally expensive step. In contrast, FM learns a vector field that defines the flow between distributions and avoids the computation of the Jacobian determinant, making it more efficient and scalable for our task. **B. Domain Identification Objective:** DECODE uses NF for domain-awareness, aiming to select the model with the highest likelihood score for making predictions. Our work, however, focuses on **OOD validation** to verify the **inputs** to motion planners.
>
> **2) Differences from SOTIF Entropy:** The entropy and energy formulas approach probability distributions in different ways. The entropy formula used in SOTIF quantifies uncertainty by evaluating the uniformity of a probability distribution. Entropy increases when probabilities are evenly distributed, indicating higher uncertainty. This makes it an effective metric for evaluating perceptual uncertainty, as it measures the uniformity of the probabilities of detected objects across categories.
>
> In our work, the energy formula (Eq.(11) in the paper) focuses on the concentration of probabilities by amplifying the influence of high-probability candidates through the use of exponential (exp) and logarithmic (log) operations. The energy formula is particularly sensitive to concentrated probability distributions where a few probabilities dominate, with higher energy values reflecting stronger confidence in the most probable outcomes. This property makes it well-suited for tasks like trajectory planning, where identifying and emphasizing the most likely trajectory is critical.
>
> **3) Our Contributions:** Our framework uniquely integrates **input validation** and **output validation** for motion planners, a design absent in both DECODE and SOTIF Entropy.
>
> **Q2: How does the identification mechanism benefit downstream planning performance, especially in closed-loop execution?**
>
> **A2:** We clarify the scope of our work while acknowledging its limitations and potential future directions.
>
> **1) Scope of REDOUBT:** REDOUBT focuses on assessing whether the vehicle's driving decisions are safe in specific scenarios. Once an unsafe condition is detected, subsequent response strategies (e.g., manual takeover or continual learning) will be applied; however, they are beyond the scope of this work. Addressing response strategies involves tackling complex problems that merit their own paper and cannot be integrated into this safety validation work within a single 9-page paper.
>
> **2) Downstream Planning Performance Improvement:** Our work concentrates on safety validation. The degree to which it can enhance downstream or overall planning performance also depends on the response strategies, such as manual takeover and continual learning. This involves tackling complex challenges like distribution shifts, designing robust learning strategies, and closed-loop training, which are outside the scope of this paper. Thus, there is currently no "entire stack" available, making it infeasible to measure downstream or overall improvements.
>
> Within the scope of safety validation, our evaluation is comprehensive and sufficient. It is worth noting that related works, such as those focused on OOD detection (Ref.[43] in the paper) or corner case detection [1], also confine the evaluation scope at a similar level.
>
> [1] Zhang, Xinhai, et al. "Finding critical scenarios for automated driving systems: A systematic mapping study." IEEE TSE 2022
>
> **Q3: How to define scenario difficulty? Distinguish hard-frequent vs.rare-trivial? Correlation between OOD cases and safety-critical failures?**
>
> **A3:** First, we follow the widely accepted NuPlan benchmark, defining a scenario as difficult if the planner's decisions: 1) result in at-fault collisions that could have been avoided, or 2) violate drivable area compliance.
>
> Secondly, our system identifies both hard-but-frequent and rare-but-'trivial' cases. Below is a table illustrating our overall performance and the breakdown of different cases. Due to space limit, we focus on the ‘recall’ metric, which measures how effectively the method covers all potential unsafe scenarios.
>
> ||PlanTF|PLUTO| GameFormer|PlanScope|Average|
> |-------|-----|------|----|---------|-----|
> |InD-Uncertain (frequent-but-hard)|0.0070|0.0126|0.0179|0.0087|0.0116|
> |OOD-Certain (rare-but-'trivial')|0.1004|0.1561|0.1050|0.0678|0.1073|
> |OOD-Uncertain (rare-and-hard)|0.7950|0.7365|0.7905|0.8237|0.7864|
> |Total|0.9024|0.9052|0.9134|0.9002|0.9053|
>
> Our overall recall of unsafe scenarios averages 0.9053. We consider three cases as unsafe: 1) OOD-Uncertain (rare-and-hard), commonly deemed unsafe by other works, covers 78.64% unsafe scenarios; 2) InD-Uncertain (frequent-but-hard) covers 1.16% unsafe scenarios. Although this may seem small, identifying such scenarios is crucial for human and property safety in real life; 3) our solution includes OOD-Certain (rare-but-'trivial') as unsafe, covers 10.73% additional unsafe scenarios. While uncertainty estimation approaches may consider this case trivial due to its high certainty, we argue that it is significant, hence our designation of rare-but-'trivial' (not really trivial).
>
> Additionally, our false positive rate is slightly lower than that of existing methods, demonstrating stronger performance in avoiding the misidentification of safe scenarios as unsafe.
>
> In summary, we consider all three cases as unsafe, and our evaluation demonstrates that each contributes to the detection of unsafe scenarios; none are unnecessary. We will include these results and discussions in the paper.
>
> **Q4: Make qualitative results more intuitive and provide code.**
>
> **A4: 1) Qualitative Results:** We will enhance the illustrations by adding a legend to make the scenarios more intuitive. We will add a legend to differentiate the elements in each scene: Orange lines for the planned trajectory of the motion planner, Blue-green lines for the trajectories of surrounding agents, and Gray lines for the candidate trajectories by the planner.
>
> **2) Detailed Analysis:** We will add a detailed analysis in **Section 4.5** to include a comprehensive discussion of failure cases observed during our experiments. **Case (b) High-Density Traffic during Left Turns:** This scenario involves a **left-turn maneuver** in a **high-density traffic environment**, which is a common challenge for motion planners. Although this category of scenario was encountered during training, we observed that the planned trajectory and the candidate trajectories still resulted in a collision with surrounding vehicles. This failure highlights the complexity of high-density interactions where subtle variations can lead to collisions despite prior training exposure. **Case (c) Unprotected Cross-Turns:** This scenario involves **unprotected cross-turns**, a category of scene that was **not observed during training**. Despite the absence of such scenarios in the training data, the planner still needs to output decisions. **Case (d) Traversing Pickup/Drop-Off Areas:** This scenario involves the **ego vehicle traversing a drop-off area** while not stopping. Such scenarios were also **not present in the training data**, and we observed that the planned trajectory drifted outside the drivable area. This failure illustrates the planner’s limitations in handling complex, context-specific scenarios that require precise spatial reasoning and awareness.
>
> **3) Code:** In the NeurIPS Submission Guidelines, open access to data and code is encouraged but not mandatory. We will release the code upon acceptance of the paper.
>
> **Q5: How is each planner trained? Are there risks of data leakage? How sensitive are results to frequency threshold?**
>
> **A5: 1) Training Details:** We provide detailed information in Appendix A.1 and A.2. To ensure consistency, all planners are trained using the original parameter settings prescribed by their respective papers or implementations.
>
> **2) Data Leakage:** There is NO data leakage in our setup. Our training and evaluation sets are derived from the nuPlan dataset's official splits, selected based on scenario frequency and type. Each scenario is assigned a unique token ID, ensuring no overlap between splits. Furthermore, the nuPlan dataset is designed to prevent leakage by not sharing data from the same day or city across splits [1].
>
> **3) Threshold of Frequency:** We chose the threshold for scenario frequency to balance the coverage of scenario types, employing a strategy similar to that used in Shifts (Ref.[33] in the paper). Specifically, we included the top 50% most frequent scenario types, which represent 95% of the total data. This threshold ensures broad coverage of diverse scenarios while maintaining a meaningful representation of the dataset.
>
> [1] Karnchanachari, Napat, et al. "Towards learning-based planning: The nuplan benchmark for real-world autonomous driving." 2024 IEEE ICRA
>
> **Q6: Ablation of verification module.**
>
> **A6:** We present this ablation in Appendix B.4. We conducted an ablation study by adjusting the hyperparameter λ (ranging from 0 to 1): When λ = 0, it represents the complete removal of the Flow Matching module. When λ = 1, it represents the complete removal of the energy-based module. The experimental results demonstrate that utilizing either module alone (λ = 0 or λ = 1) does not achieve optimal performance. This demonstrates that both are critical for performance improvement.

---

> > ### Comment · Reviewer_pm9A · 2025-08-04
> >
> > After a detailed analysis through the rebuttal, the reviewer is convinced by the authors regarding the novelty in designing the safety validation framework. Still, its current evaluation is limited (as also noted by other reviewers) without close-loop planners. Hence, the reviewer is inclined to raise as the borderline score.

---

> > > ### Author Response · Authors · 2025-08-06
> > >
> > > Thanks for your comment and for acknowledging the strengths of our paper. We want to provide a further explanation that helps resolve your concerns.
> > >
> > > (1) The new results presented in our rebuttal (specifically, the Recall metric) show that our method not only identifies unsafe cases previously missed but also has lower false positives, meaning that it would not unnecessarily report safe cases as unsafe ones. For further details, please also refer to our response A1 to Reviewer Kf2E.
> > >
> > > (2) We have already noted that related works, such as those on OOD detection [1] and corner case detection [2], limit their evaluations within the same scope as ours. Even so, we can meet the reviewer's demand by proposing a simple "full-stack," closed-loop evaluation strategy: whenever our approach detects an unsafe case, a manual takeover is triggered. Assuming manual driving is perfectly safe, this "full-stack" evaluation naturally leads to improved safety outcomes based on the results in (1). Of course, if one argues that manual takeover is not entirely safe, then no solution can be conclusively proven superior under such an assumption.
> > >
> > > [1] Yang, Jingkang, et al. "Generalized out-of-distribution detection: A survey." IJCV 2024
> > >
> > > [2] Zhang, Xinhai, et al. "Finding critical scenarios for automated driving systems: A systematic mapping study." IEEE TSE 2022

---

### Official Review · Reviewer_g3mR · 2025-07-02

**Clarity:** 3
**Significance:** 2
**Originality:** 3
**Rating:** 2
**Confidence:** 4

**Summary:**

The paper presents REDOUBT, a novel approach for self-driving ML planner’s OOD detection. In addition to the trajectory decoding head, it adds two head that predicts 1) uncertainty, supervised by heuristics on collision and offroads and 2) a flow-matching based OOD detector.

**Questions:**

- Given the OOD detection enabled by REDOUBT, how much better does the overall planning stack improve? Can the authors evaluate this with standard planning metrics?
- It’s unclear to me why it has to be flow-matching which requires solving ODE during inference if the sole purpose is to score the trajectory prediction. For example, why not a conditional normalizing flow or VAE?
- I’d like to see some case or qualitative analysis on the failure modes, e.g. false positives and negatives.
- What are the additional computes/latency of REDOUBT to a planner stack?

**Ethical Concerns:**

["NO or VERY MINOR ethics concerns only"]

**Final Justification:**

After reading the authors' rebuttal, I think two major concerns remain:

1) The design justification for using flow matching (A3) is a bit hand-wavy. A quantitative comparison would have been nice here.
2) Motion planning from structured inputs is already a self-contained task, and if the method is evaluated without its full closed-loop metrics, I am afraid the scope of the paper becomes way too niche. In addition, without the full metrics the claim that "the proposed method can be plugged in any ML planner" is not sufficiently supported, because it's unclear how a ML planner work when it plugs in in the proposed method.

Based on the above, I am unable to recommend acceptance at this point.

**Limitations:**

The main limitations at the moment is the lack of closed-loop evaluation with planning metrics on the entire system. Also, no explicit discussion on the limitations.

**Quality:**

2

**Strengths And Weaknesses:**

### Strengths
+ The paper is well written and tackles an important problem that is relatively under-explored.
+ The proposed method is intuitive and easily integratable to arbitrary (ML)-planner stack.


### Weaknesses
- The statement in L57 “We propose REDOUBT, the first systematic safety validation framework for autonomous vehicle motion planning” felt overclaiming. There has been thorough prior work on self-driving safety validation [1,2]. I’d make sure the claim is more specific, e.g. clarify that the approach focuses on OOD detection.
- My main concern is that there is no closed-loop evaluation on the entire stack – given the OOD detection enabled by REDOUBT, how much does the overall planning stack improve?

[1] Determining Absence of Unreasonable Risk: Approval Guidelines for an Automated Driving System Deployment, Favaro et al.
[2] Comparison of Waymo rider-only crash data to human benchmarks at 7.1 million miles, Kusano et al.

---

> ### Author Rebuttal · Authors · 2025-07-31
>
> Thank you for reviewing our paper. We appreciate your valuable feedback and will try to address your concerns below:
>
> **Q1: Clarify the statement in L57 and make it more specific.**
>
> **A1:** Thank you for pointing this out. We agree that the statement in L57 could be more specific. In the revised version, we will reframe the claim as follows: *"We propose REDOUBT, the first framework to systematically integrate input inspection through out-of-distribution (OOD) detection and output inspection via uncertainty estimation for the safety validation of autonomous vehicle motion planning."*
>
> **Q2: The closed-loop evaluation of the planning stack and how the OOD detection enabled by REDOUBT improves overall planning performance?**
>
> **A2:** We appreciate the reviewer’s insightful question regarding the closed-loop evaluation of the entire planning stack and how the OOD detection enabled by **REDOUBT** improves overall planning performance. We would like to **clarify the scope of our work while acknowledging its limitations and potential future directions**.
>
> **1) Scope of REDOUBT:** The primary focus of REDOUBT is safety validation, which assesses whether the vehicle's driving decisions are safe in specific scenarios. Once an unsafe condition is detected, subsequent response strategies (e.g., manual takeover to ensure safety or continual learning for model improvement) will be applied; however, they are beyond the scope of this work. Addressing response strategies involves tackling substantial and complex problems that merit their own paper and cannot be integrated into this safety validation work within a single 9-page paper. We intend to explore these strategies in future work.
>
> **2) Why Closed-Loop Evaluation of Overall Planning Stack Is Ill-Suited for Our Work:** As mentioned, this work concentrates solely on safety validation. The degree to which it can enhance overall planning performance also depends on the employed response strategies, such as manual takeover and continual learning. This involves tackling complex challenges like distribution shifts, designing robust learning strategies, and closed-loop training, which are outside the scope of this paper. Therefore, there is currently no "entire stack" available, making it infeasible to measure overall improvements.
>
> Within the scope of safety validation, we believe our evaluation is comprehensive and sufficient. It is worth noting that related works, such as those focused on OOD detection [1] or corner case detection [2], also confine their evaluation scope at a similar level.
>
> [1] Yang, Jingkang, et al. "Generalized out-of-distribution detection: A survey." IJCV 2024
>
> [2] Zhang, Xinhai, et al. "Finding critical scenarios for automated driving systems: A systematic mapping study." IEEE TSE 2022
>
> **Q3: Why choose flow-matching for inference instead of a conditional normalizing flow or VAE?**
>
> **A3: 1) Continuous-Time Trajectory Modeling:** Flow Matching is specifically designed for modeling continuous-time stochastic processes. By formulating the transformation as an ODE, Flow Matching can naturally represent the fine-grained, continuous dynamics of real-world trajectories.
>
> **2) Comparison with Conditional Normalizing Flows (CNF):** While CNFs are powerful, they require computation of the Jacobian determinant during training and inference, which becomes costly as data dimensionality grows—a common scenario in trajectory prediction. In contrast, Flow Matching sidesteps this bottleneck by not requiring Jacobian computations, making it more scalable and efficient for high-dimensional, complex trajectories.
>
> **3) Limitations of VAEs:** VAEs model the latent space by assuming a simple prior distribution (usually a standard Gaussian). This assumption can limit their ability to accurately capture the full diversity and complexity present in real-world trajectory data, often resulting in less accurate or less diverse predictions.
>
> **Q4: I’d like to see some case or qualitative analysis on the failure modes, e.g. false positives and negatives.**
>
> **A4:** Section 4.5 presents trajectory planning scenarios of PlanScope recorded during closed-loop reactive simulation experiments. We will provide a more detailed analysis, particularly for the failure cases shown in Fig. 3b and 3d as follows:
>
> We identified and analyzed several representative failure cases encountered in our experiments: **Case (b) High-Density Traffic during Left Turns.** This scenario involves a left-turn maneuver in a high-density traffic environment, which is a common challenge for motion planners. Although this category of scenario was encountered during training, we observed that the planned trajectory by PlanScope planner and the candidate trajectories still resulted in a collision with surrounding vehicles. This failure highlights the complexity of high-density interactions where subtle variations can lead to collisions despite prior training exposure. **Case (c) Unprotected Cross-Turns.** This scenario involves unprotected cross-turns, a category of scene that was not observed during training. Despite the absence of such scenarios in the training data, the planner still needs to output decisions. **Case (d) Traversing Pickup/Drop-Off Areas.** This scenario involves the ego vehicle traversing a drop-off area while not stopping. Such scenarios were also not present in the training data, and we observed that the planned trajectory drifted outside the drivable area. This failure illustrates the planner’s limitations in handling complex, context-specific scenarios that require precise spatial reasoning and awareness.
>
> **Q5: What are the additional computes/latency of REDOUBT to a planner stack?**
>
> **A5:** Due to page limitations, we present the inference times of our method as applied to various motion planners in Appendix C, Table 8. REDOUBT introduces an additional inference time of only 3.12 to 10.79 ms. Considering that the original planner operates at 20 Hz (resulting in a latency of 50 ms per frame), the extra latency from REDOUBT is minimal. Importantly, the total latency remains well below the maximum threshold of 1000 ms per inference mandated by the nuPlan benchmark. These results demonstrate that REDOUBT is computationally efficient and capable of meeting the real-time performance requirements of modern motion planners. Based on your feedback, we will expand the discussion in Appendix C to provide a more comprehensive analysis of inference times and their real-time applicability.

---

> ### Comment · Reviewer_g3mR · 2025-08-05
>
> Thank you for the response. After reading the authors' rebuttal, I think two major concerns remain:
> - The design justification for using flow matching (A3) is a bit hand-wavy. A quantitative comparison would have been nice here.
> - Motion planning from structured inputs is already a self-contained task, and if the method is evaluated without its full closed-loop metrics, I am afraid the scope of the paper becomes way too niche. In addition, without the full metrics the claim that "the proposed method can be plugged in any ML planner" is not sufficiently supported, because it's unclear how a ML planner work when it does plugs in in the proposed method.
>
> Based on the above, I am unable to recommend acceptance at this point.

---

> > ### Author Response · Authors · 2025-08-06
> >
> > We sincerely appreciate the reviewer's thoughtful feedback. We respectfully request the reviewer to reconsider the recommendation in light of the following factual evidence.
> >
> > **(1).** There is numerical evidence showing that normalizing flows are computationally heavy, e.g., in [1] [2] below, learning likelihoods with normalizing flows involves significantly higher computational costs during both training and inference—approximately 6–10 times slower compared to flow matching. Conditional normalizing flows (CNFs) have even higher computational cost than standard (unconditional) normalizing flows due to the additional conditioning step that integrates extra information into the flow transformations[3].
> >
> > [1] Leon, et al. "Equivariant Flow Matching." NeurIPS 2023
> >
> > [2] Ben-Hamu, et al. “Matching Normalizing Flows and Probability Paths on Manifolds”  ICML 2022
> >
> > [3] Papamakarios, et al. "Normalizing flows for probabilistic modeling and inference." JMLR 2021
> >
> > **(2).** The new results presented in our rebuttal (specifically, the Recall metric) show that our method not only identifies unsafe cases previously missed but also has lower false positives, meaning that it would not unnecessarily report safe cases as unsafe ones. For further details, please also refer to our response A1 to Reviewer Kf2E.
> >
> > **(3).** We have already noted that related works, such as those on OOD detection [1] and corner case detection [2], limit their evaluations within the same scope as ours. Even so, we can meet the reviewer's demand by proposing a simple "full-stack," closed-loop evaluation strategy: whenever our approach detects an unsafe case, a manual takeover is triggered. Assuming manual driving is perfectly safe, this "full-stack" evaluation naturally leads to improved safety outcomes based on the results in (2). Of course, if one argues that manual takeover is not entirely safe, then no solution can be conclusively proven superior under such an assumption.
> >
> > [1] Yang, Jingkang, et al. "Generalized out-of-distribution detection: A survey." IJCV 2024
> >
> > [2] Zhang, Xinhai, et al. "Finding critical scenarios for automated driving systems: A systematic mapping study." IEEE TSE 2022
> >
> > As authors, our responsibility is to present accurate and objective information to the best of our ability. We trust that the reviewers will evaluate the submission based on these facts and make a fair recommendation grounded in evidence rather than subjective impressions.

---

> ### Comment · Reviewer_g3mR · 2025-08-08
>
> Thank you for your additional response. My concerns remain:
>
> > There is numerical evidence showing that normalizing flows are computationally heavy, e.g., in [1] [2] below
>
> Thanks for linking the external references. This does not change the fact that the paper does not quantitatively justify flow matching as a core design choice, which is my original critique. The paper does not demonstrate that using flow-matching as a generic density model, is superior / more efficient compared to alternatives on the **target setup**. Simply stating it's faster than CNF does not feel like sufficient support.
>
> > We have already noted that related works, such as those on OOD detection [1] and corner case detection [2], limit their > evaluations within the same scope as ours.
> > [1] Yang, Jingkang, et al. "Generalized out-of-distribution detection: A survey." IJCV 2024
> >
> >[2] Zhang, Xinhai, et al. "Finding critical scenarios for automated driving systems: A systematic mapping study." IEEE TSE 2022
>
> [1] is a survey paper, and [2] is a systematic mapping study. [2] is also for a different domain. Neither of these papers claims a contribution that would require the motion planning validation expected here. Overall, not having planning metrics for a method, that limits its scope for validating only the motion planner, remains my main concern. I don't think having closed-loop motion planning metrics on planners is beyond the scope, especially considering the authors have already integrated the proposed method to different planners (Section 4).

---

> > ### Author Response · Authors · 2025-08-08
> >
> > Thanks for your feedback.
> >
> > > The paper does not demonstrate that using flow-matching as a generic density model, is superior / more efficient compared to alternatives on the target setup.
> >
> > We did not fully understand your logic here. You challenged flow matching and suggested alternatives. We clearly demonstrated that your proposed alternative is computationally intensive, as supported by numerical evidence. There are tons of models here; no matter which one we use, one can always suggest a different one and requires quantitative justification.
> >
> > > I don't think having closed-loop motion planning metrics on planners is beyond the scope.
> >
> > There might be a misunderstanding here. First, we follow the widely accepted NuPlan benchmark, defining a scenario as unsafe based on two commonly used closed-loop motion planning metrics: 1) if the planner's decisions result in at-fault collisions that could have been avoided, or 2) violate drivable area compliance. Then, we clarified that our results (particularly the recall) are already sufficient to support the stronger safety claim, as a higher recall indicates that more unsafe scenarios are detected to avoid collisions and compliance violations. Once an unsafe condition is detected, how subsequent response strategies (e.g., manual takeover or continual learning) will be applied is beyond the scope of this work. Nevertheless, if we have to give a full-stack, closed-loop evaluation, we can suggest one: whenever our approach detects an unsafe case, a manual takeover is triggered. By assuming manual driving is perfectly safe, we will get improved final safety outcomes based on the recall results. In the revised paper, we can use this naive strategy as an example scenario to report the full-stack, closed-loop evaluation.

---

> ### Comment · Reviewer_g3mR · 2025-08-08
>
> > We did not fully understand your logic here. You challenged flow matching and suggested alternatives. We clearly demonstrated that your proposed alternative is computationally intensive, as supported by numerical evidence. There are tons of models here; no matter which one we use, one can always suggest a different one and requires quantitative justification.
>
> My critique was that the paper did not demonstrate why its choice of flow-matching is superior for this task compared to alternatives for density estimation. The authors' rebuttal did not convince me because 1. **External evidence does not substitute the empirical, task-specific comparison**. An ablation study comparing against baselines on the same setup would resolved the concerns.  2. I think **requesting a baseline comparison on a core design choice is not an open-ended demand**.
>
> > First, we follow the widely accepted NuPlan benchmark, defining a scenario as unsafe based on two commonly used closed-loop motion planning metrics: 1) if the planner's decisions result in at-fault collisions that could have been avoided, or 2) violate drivable area compliance. Then, we clarified that our results (particularly the recall) are already sufficient to support the stronger safety claim, as a higher recall indicates that more unsafe scenarios are detected to avoid collisions and compliance violations.
>
> If I understand correctly, the nuPlan metrics (collisions, drivable area violations) are to *label* an unsafe scenario. This is not a closed-loop evaluation. A closed-loop evaluation measures the **end-to-end performance of a planner integrated with their module**, reporting the final collision rate and other violations on the benchmark. The paper does not provide this.
>
> > Nevertheless, if we have to give a full-stack, closed-loop evaluation, we can suggest one: whenever our approach detects an unsafe case, a manual takeover is triggered. By assuming manual driving is perfectly safe, we will get improved final safety outcomes based on the recall results.
>
> The reviewer is looking for **standard end-to-end nuPlan metrics, evaluating planners with the proposed module plugged in**. The fallback planner need not be perfect, and the integrated planner need not to improve planning scores all around, but improving on safety related measure is sufficient to demonstrate the efficacy. This is not beyond the scope since the authors have already integrated the proposed module with other planners. The authors' proposed evaluation sidesteps the established evaluation protocol. If the fallback driver is an oracle that does perfect driving, that of course the overall system is better no matter how the OOD system works -- the system is better even if the OOD detector constantly reports positives. This does not tell the audience about the practical utility or failure modes of the proposed approach.
>
> The expectation for this paper, which makes safety claims for motion planning, is to demonstrate a quantitative improvement on standard, widely accepted planning benchmark metrics. I do not think this request is questionable at the moment.

---

> > ### Author Response · Authors · 2025-08-09
> >
> > Thanks for your feedback.
> >
> > >  My critique was that the paper did not demonstrate why its choice of flow-matching is superior for this task compared to alternatives for density estimation. The authors' rebuttal did not convince me because 1. External evidence does not substitute the empirical, task-specific comparison. An ablation study comparing against baselines on the same setup would resolved the concerns. 2. I think requesting a baseline comparison on a core design choice is not an open-ended demand.
> >
> > Response: We think the reviewer's request is open-ended: 1) Sufficient Baselines Already Provided: Our results (Table 1) show that our OOD solution outperforms various existing OOD methods. 2) Unfair Cross-Domain Comparison Demands: The reviewer specified models from other domains that are not baselines, and required us to adapt them to the OOD context as baselines for comparisons. This expectation adds complexity without clear justification. 3) Endless Cycle of Critique: With the vast number of cross-domain models available, any choice we make could lead to further requests for additional comparisons, creating an endless cycle of open-ended demands.
> >
> > >  If I understand correctly, the nuPlan metrics (collisions, drivable area violations) are to label an unsafe scenario. This is not a closed-loop evaluation. A closed-loop evaluation measures the end-to-end performance of a planner integrated with their module, reporting the final collision rate and other violations on the benchmark. The paper does not provide this.
> >
> > Response: The reviewer has misunderstood the concepts. All our results are evaluated in a closed-loop manner: We integrated our module into various planners, let the “planner” actually control the ego vehicle in the nuPlan benchmark, and computed the nuPlan metrics (collisions, drivable area violations)  for each scenario over the generated trajectory. Scenarios are classified as safe or unsafe based on the collisions and drivable area violations observed in that closed-loop rollout, from which the recall metric is derived. Our work improves this closed-loop end-to-end recall by 25-35% compared to existing methods. "For further details, please also refer to our response A1 to Reviewer Kf2E", as we noted in our earlier discussion.
> >
> > >  If the fallback driver is an oracle that does perfect driving, that of course the overall system is better no matter how the OOD system works -- the system is better even if the OOD detector constantly reports positives. This does not tell the audience about the practical utility or failure modes of the proposed approach.
> >
> > Response: The reviewer expresses concerns regarding the false positive rate (FPR) of our approach. However, our original paper (Tables 1-4) and our rebuttal clearly highlight that "our false positive rate is lower than that of existing methods, demonstrating stronger performance in avoiding the misidentification of safe scenarios as unsafe." The reviewer has overlooked these important points.

---

> ### Comment · Reviewer_g3mR · 2025-08-09
>
> Thanks for the additional response. The authors' latest rebuttal misinterprets my critiques and does not address the aforementioned flaws. My current decision on the paper is based on the following issues.
> - On justifying the design choice: the authors claim that requesting ablation study on flow-matching for density estimation is _open-ended_ and _unfair_. I respectively disagree for the following reasons.
>   - In the latest rebuttal, the authors conflate baselines in OOD scoring (table 1) with the underlying generative models that provide the estimation in REDOUBT. My critique is regarding the lack of the ablation study that justify the design choice of using flow-matching. I think the proper baselines here are other well-established generative models such as VAEs and CNFs, in place of flow-matching in the REDOUBT framework, as mentioned in the earlier response. I.e, the comparison would be "REDOUBT (flow-matching) vs REDOUBT (VAE)", rather than "REDOUBT vs other OOD methods".
>   - I do not agree with the authors that VAEs and CNFs are cross-domain models, but they are standard, general-purpose approaches for generative modeling.
>   - Similarly I do not think comparing to one, or two aforementioned ablation baseline is "open-ended".
> - On evaluation: the authors insist their evaluation is "closed-loop" and that I have misunderstood. I believe the authors might be conflating the OOD classification task, whose inputs and labels are indeed generated in closed-loop rollouts, with the true system-level planning evaluations. The latter requires integrating REDOUBT with planners (the authors already did this) and a fallback policy, and demonstrate that the infraction rates are lower with REDOUBT enhanced planners. This is not the same as reporting classification metrics like recall.
>
> Additional I would like to clarify on "the oracle driver argument" in my last response. My argument on the oracle driver is to demonstrate why the authors' proposed experiment does not equate the full planning evaluations. A perfect oracle makes the trade-off between false positives and false negatives meaningless. The authors response, that the false positive rate of REDOUBT is lower than that of the OOD detection baselines, does not refute the fact **the proposed experiment cannot substitute full planning evaluation**.

---

### Official Review · Reviewer_Kf2E · 2025-07-03

**Clarity:** 3
**Significance:** 3
**Originality:** 3
**Rating:** 4
**Confidence:** 2

**Summary:**

This paper introduces REDOUBT, a novel duo safety validation framework for autonomous driving motion planning that jointly considers scenario distribution (via OOD detection) and output uncertainty (via planning confidence). The framework is validated across multiple learning-based motion planners on the nuPlan dataset under both open-loop and closed-loop settings and demonstrates consistent improvements over existing baselines.

**Questions:**

* The paper evaluates OOD detection and uncertainty estimation separately, but lacks evaluation for overall validation effectiveness. Can the authors discuss how to evaluate the performance of the whole system?
* How can REDOUBT be applied in end-to-end autonomy systems that operate directly on raw sensor inputs

**Ethical Concerns:**

["NO or VERY MINOR ethics concerns only"]

**Limitations:**

yes

**Quality:**

3

**Strengths And Weaknesses:**

**Strengths**
- The duo validation perspective is insightful. This paper provides a compelling argument for jointly validating both input distribution and output uncertainty, highlighting overlooked unsafe cases that each dimension alone would miss (InD + Uncertain and OOD + Certain). This insight is practically meaningful for AV safety.
- The use of flow matching for OOD likelihood estimation is interesting and shows consistent improvements over a wide range of post hoc and feature-space baselines.
- The decision uncertainty estimation module works for both open-loop and closed-loop settings, and is evaluated across multiple motion planners and under three settings (OP/CNR/CR), with consistent performance gains.

**Weaknesses**
- The paper evaluates OOD detection and uncertainty estimation independently but does not quantify REDOUBT’s overall safety validation performance as a unified system. It would be valuable to define and report metrics that directly reflect end-to-end validation effectiveness.
- The method assumes access to structured scenario inputs (e.g., object trajectories, maps), which may not be available in end-to-end autonomous systems that go directly from raw sensor input to planned trajectory. Can REDOUBT be adapted to operate in such sensor-to-plan pipelines?

---

> ### Author Rebuttal · Authors · 2025-07-31
>
> Thank you for reviewing our paper. We appreciate your valuable feedback and will try to address your concerns below:
>
> **Q1: Discussion on metrics and evaluation for REDOUBT’s unified end-to-end safety validation.**
>
> **A1:** Thank you for your feedback. Based on your comment, we present the validation performance of the entire REDOUBT system when OOD detection and uncertainty estimation are used together. Due to space constraints, we focus on the ‘recall’ metric, which measures how effectively the method covers all potential unsafe scenarios.
>
> | Recall           | PlanTF | PLUTO  | GameFormer | PlanScope | Average |
> | ---------------- | ------ | ------ | ---------- | --------- | ------- |
> | Uncertain (lGMM) | 0.5350 | 0.6588 | 0.5181     | 0.4995    | 0.5529  |
> | OOD (Energy)     | 0.5952 | 0.6582 | 0.6497     | 0.7261    | 0.6573  |
> | Ours             | 0.9024 | 0.9052 | 0.9134     | 0.9002    | 0.9053  |
>
> Our results show a significant increase in recall, rising from 0.5529 for SOTA uncertainty estimation in our baselines and 0.6573 for OOD detection, to 0.9053 with our approach. This demonstrates our method's effectiveness in identifying a broader range of unsafe scenarios.
>
> Additionally, it is worth noting that our false positive rate is slightly lower than that of existing methods, demonstrating stronger performance in avoiding the misidentification of safe scenarios as unsafe. We will incorporate these results and discussions in the paper.
>
> **Q2: How can REDOUBT be applied in end-to-end autonomy systems that operate directly on raw sensor inputs?**
>
> **A2:** Thank you for your comment. Yes, REDOUBT applies to end-to-end sensor-to-plan pipelines. Inside most modern end-to-end architectures [1], raw sensor inputs (e.g., maps, cameras, lidars) are first encoded into a latent scene representation using neural networks, and then trajectory planning is conducted in this latent space.
>
> **Scenario inputs:** As illustrated in Figure 1, our system operates on the outputs of the scene encoder (i.e., the latent representation) rather than the raw inputs. This alignment makes it well-suited for end-to-end autonomous systems. Based on your feedback, we will refine Figure 1 and the accompanying text to further clarify this point.
>
> [1] Lan, Gongjin, and Qi Hao. "End-to-end planning of autonomous driving in industry and academia: 2022-2023." arXiv preprint 2023.

---

### Official Review · Reviewer_GyUM · 2025-07-22

**Clarity:** 3
**Significance:** 3
**Originality:** 3
**Rating:** 4
**Confidence:** 4

**Summary:**

This paper presents REDOUBT, a safety validation framework for autonomous vehicle motion planning that addresses gaps in current safety assessment approaches. The key innovation lies in its dual validation mechanism that simultaneously evaluates both input distributions (traffic scenarios) and output uncertainty (planning decisions). The framework tackles two challenges: Out-of-Distribution (OOD) detection for identifying unusual driving scenarios, and decision uncertainty estimation for assessing the reliability of planning outputs. REDOUBT employs latent flow matching for OOD detection and an energy-based method for uncertainty quantification based on safety risks from obstacle proximity. The authors demonstrate through comprehensive experiments on the nuPlan dataset that their approach significantly outperforms existing methods across multiple evaluation metrics, providing enhanced safety guarantees for autonomous driving systems.

**Questions:**

See the weaknesses.

**Ethical Concerns:**

["NO or VERY MINOR ethics concerns only"]

**Final Justification:**

Just keep my original rating.

**Limitations:**

yes

**Quality:**

3

**Strengths And Weaknesses:**

Strengths
- The paper tries to address a critical and practically important problem in autonomous driving.
- The proposed approach is technically sound and well-motivated.
- Comprehensive experimental evaluation on the nuPlan dataset with comparison against existing methods shows clear performance improvements across multiple scenarios.

Weaknesses
- Limited discussion of computational overhead and real-time performance implications.
- While the approach shows promise, the practical deployment challenges and integration with existing autonomous driving stacks are not thoroughly discussed.
- More detailed analysis of failure cases would be valuable.
- Only InD scenarios with certain decisions can be considered safe; such a condition may be too strict and could cause excessive takeovers in practical driving scenarios.

---

> ### Author Rebuttal · Authors · 2025-07-31
>
> Thank you for reviewing our paper. We appreciate your valuable feedback and will try to address your concerns below:
>
> **Q1: Discussion of computational overhead and real-time performance implications.**
>
> **A1:** Due to the page limit, we report the inference times of our method with different motion planners in Appendix C, Table 8. REDOUBT introduces an additional inference time of only 3.12 to 10.79 ms. Considering that the original planner operates at 20 Hz (resulting in a latency of 50 ms per frame), the extra latency from REDOUBT is minimal. Importantly, the total latency remains well below the maximum threshold of 1000 ms per inference mandated by the nuPlan benchmark. These results demonstrate that REDOUBT is computationally efficient and capable of meeting the real-time performance requirements of modern motion planners. Based on your feedback, we will expand the discussion in Appendix C to provide a more comprehensive analysis of inference times and their real-time applicability.
>
> **Q2: Discussion on practical deployment challenges and integration with autonomous driving stacks.**
>
> **A2:** Based on your feedback, we will expand our discussion to address our approach's practical deployment and integration with existing autonomous driving stacks as follows: Notably, several leading autonomous driving companies, such as Momenta [1], have already adopted learning-based planners in their systems. Our method is well-suited for such deployments, as it integrates seamlessly into the pipeline. Specifically, the input to our method would be the latent space representation from the autonomous driving stack, and the output would include whether it represents an out-of-distribution (OOD) scenario and risk probabilities of potential driving violations. Several practical challenges need to be addressed for real-world deployment. Determining takeover frequency: Our approach could identify more unsafe scenarios than traditional methods. This raises the question of how to balance sensitivity and specificity in decision-making. To address these challenges, we propose that the output of our method serves as helpful, supplementary information to existing decision-making mechanisms within the stack. For instance, it could integrate with existing systems to trigger takeover requests as needed. Additionally, our system could be optionally enabled to prioritize safety in specific scenarios, such as high-risk environments or areas prone to OOD events.
>
> [1] Lan, Gongjin, and Qi Hao. "End-to-end planning of autonomous driving in industry and academia: 2022-2023." arXiv preprint 2023.
>
> **Q3: More detailed analysis of failure cases.**
>
> **A3:** To address this, we will add more detailed analysis in **Section 4.5** to include a comprehensive discussion of failure cases observed during our experiments as follows: We identified and analyzed several representative failure cases encountered in our experiments: **Case (b) High-Density Traffic during Left Turns:** This scenario involves a left-turn maneuver in a high-density traffic environment, which is a common challenge for motion planners. Although this category of scenario was encountered during training, we observed that the planned trajectory by PlanScope planner and the candidate trajectories still resulted in a collision with surrounding vehicles. This failure highlights the complexity of high-density interactions where subtle variations can lead to collisions despite prior training exposure. **Case (c) Unprotected Cross-Turns:** This scenario involves unprotected cross-turns, a category of scene that was not observed during training. Despite the absence of such scenarios in the training data, the planner still needs to output decisions. **Case (d) Traversing Pickup/Drop-Off Areas:** This scenario involves the ego vehicle traversing a drop-off area while not stopping. Such scenarios were also not present in the training data, and we observed that the planned trajectory drifted outside the drivable area. This failure illustrates the planner’s limitations in handling complex, context-specific scenarios that require precise spatial reasoning and awareness.
>
> **Q4: Strict InD conditions may lead to excessive takeovers in practical driving.**
>
> **A4:** We acknowledge that our approach MIGHT identify more scenarios as unsafe. However, it is worth noting that our false positive rate is slightly lower than that of existing methods, demonstrating stronger performance in avoiding the misidentification of safe scenarios as unsafe. Furthermore, the benefit of our research is justified as follows. First, our approach can benefit offline continual learning by providing a more comprehensive set of unsafe scenarios for model improvement. Second, in online driving, we do not advocate for automatic manual takeovers in every reported unsafe situation. Instead, the output of our method serves as supplementary information that supports other decision-making mechanisms in initiating takeover requests. It can also be optionally activated in scenarios that empirically warrant heightened caution—such as detecting a road construction flag or a school bus with children ahead. Compared to existing approaches that may overlook potentially unsafe situations, our caution-oriented method offers a safer alternative. After all, when human life is at stake, it is better to err on the side of caution.

---

### Note · Authors · 2025-08-13

We sincerely thank all reviewers for their thoughtful and constructive feedback.

Our work REDOUBT tackles a critical, underexplored problem: end-to-end validation for motion planners that integrates both input and output validation. The REDOUBT framework is intuitive and easy to plug into existing ML-based planning stacks. Extensive experiments compare against OOD baselines in both open-loop and closed-loop settings, demonstrating consistent gains across widely used planning algorithms. All Reviewers praised the novelty, practicality, and effectiveness of our method.

We responded to all the reviewers' questions point by point, with in-depth discussion and necessary additional experiments included in the rebuttal. Here are several key concerns raised by the reviewers and our corresponding responses:

**(1). Discussion of computational overhead**. In our responses to reviewers GyUM, g3mR, and pm9A, we expanded Appendix C to provide a more comprehensive analysis of inference times and their real-time applicability.

**(2). Discussion on integration with autonomous driving stacks**. In our responses to reviewers GyUM and Kf2E, we expanded our discussion to address practical deployment and integration with existing autonomous driving stacks. We also explain how REDOUBT applies to end-to-end sensor-to-plan pipelines by operating on the latent representation.

**(3). More detailed analysis of failure cases**. In our responses to reviewers GyUM, g3mR, and pm9A, we added a more thorough study in Section 4.5, including a comprehensive discussion of failure cases observed during our experiments.

**(4). Design choice of verification modules**

1. Comparison with prior work: As suggested by Reviewer pm9A, we included a detailed comparison with DECODE and SOTIF Entropy.
2. Module importance (ablation): Appendix B.4 reports complete removal of the Flow Matching and energy-based modules.

**(5). Evaluation of REDOUBT's unified end-to-end safety validation**. We added new rebuttal results (Recall) showing that REDOUBT detects unsafe cases missed by prior methods while reducing false positives, thereby avoiding unnecessary flags on safe scenarios. We also discussed a full-stack, closed-loop evaluation strategy: when REDOUBT flags an unsafe case, the system triggers a manual takeover.

---

### Decision · Program_Chairs · 2025-09-17

**Decision:**

Accept (poster)

**Comment:**

This paper received ratings of 4442. The authors propose a new safety validation framework for autonomous vehicle motion planning that integrates out-of-distribution detection and uncertainty estimation. The reviewers agree that the authors are addressing a critical issue in autonomous driving and have conducted thorough experimental validation across various planners. The rebuttal has addressed computational overhead concerns and practical deployment challenges by expanding discussions and providing additional analyses in the rebuttal.

During the discussion period, there were still some unresolved issues regarding the use of flow matching and the interpretation of closed-loop evaluation from Reviewer g3mR. The AC has gone through the details and believes that it is reasonable to use flow matching based on the author's argument. The authors are still suggested to investigate using VAE or CNF as the baselines and provide some comparisons, but the AC does not think this is a critical matter. Another issue is the closed-loop evaluation. The AC understands the concerns of the Reviewer g3mR because the experiments are not conducted in a full-stack closed-loop manner, but the closed-loop metrics used in the submission are commonly used in the benchmark nuPlan, which is a closed-loop ML-based planning benchmark for autonomous vehicles. The AC suggests that the authors provide more information about these metrics.

Overall, the paper proposes a practical and systematic safety validation framework for autonomous driving, and the AC believes the work can be accepted for NeurIPS 2025.